# VISION-LANGUAGE MODELS UNLOCK TASK-CENTRIC LATENT ACTIONS

## ABSTRACT

Latent Action Models (LAMs) have rapidly gained traction as an important component in the pre-training pipelines of leading Vision-Language-Action models. However, they fail when observations contain action-correlated distractors, often encoding noise instead of meaningful latent actions. Humans, on the other hand, can effortlessly distinguish task-relevant motions from irrelevant details in any video given only a brief task description. In this work, we propose to utilize the common-sense reasoning abilities of Vision-Language Models (VLMs) to provide promptable representations, effectively separating controllable changes from the noise in unsupervised way. We use these representations as targets during LAM training and benchmark a wide variety of popular VLMs, revealing substantial variation in the quality of promptable representations as well as their robustness to different prompts and hyperparameters. Interestingly, we find that more recent VLMs may perform worse than older ones. Finally, we show that simply asking VLMs to ignore distractors can substantially improve latent action quality, yielding up to a six-fold increase in downstream success rates on Distracting MetaWorld.

## 1 INTRODUCTION

Latent action models [42, 53] have quickly become integral to the pre-training pipelines of leading Vision–Language–Action (VLA) systems [5, 7, 58, 6, 25]. By inferring compact, semantically meaningful latent action representations at scale, Latent Action Models (LAM) mitigate the scarcity of high-quality action-labeled data, giving a promise to unlock vast amounts of unlabeled videos [33]. Removing the data bottleneck facilitates further scaling in embodied AI and robotics; consequently, any improvements to LAMs can have outsized downstream impacts.

Unfortunately, most LAMs [42, 53, 10, 18] to date have been trained on relatively clean datasets, where changes between observations can be explained almost entirely by ground-truth actions—such as in static scenes with a single manipulator. In contrast, real-world data often contains numerous action-correlated distractors, including background human movement or other spurious correlations. As shown by Nikulin et al. [36], Zhang et al. [56], without explicit supervision, LAMs struggle to disentangle controllable changes from noise, completely failing to produce meaningful latent actions in the presence of action-correlated distractors. While providing supervision via true actions can be effective [36], it is not scalable — especially in domains where these actions are impossible to obtain, such as egocentric human videos.

Humans, however, interpret the world through semantics rather than raw pixels, and with only a brief task description can easily separate task-relevant features from irrelevant details in any video. Wouldn't it also be convenient to simply ask LAM to focus on the relevant features, e.g. robotic arm, and ignore any other details? Inspired by the work of Chen et al. [9], Huang et al. [24] on promptable representations, we propose to utilize the common-sense reasoning abilities of modern Vision-Language Models (VLMs) as an unsupervised approach for effectively separating controllable changes from noise, thereby restoring the LAM's ability to recover ground-truth actions even in the presence of distractors.

In this work, we present our investigation on whether promptable representations produced by prompting VLMs to focus on task-specific details can serve as an effective target for latent action learning in the presence of distractors. Using Distracting MetaWorld as our main environment (Section 3), we begin from a simple demonstration experiment, showing that limitations of LAM

can be mitigated with the right target (Section 4). We then conduct large-scale benchmarking of different VLMs, comprising over 29k+ experiments, to assess their effectiveness at providing promptable representations (Section 5), revealing substantial variation in quality and robustness to hyperparameters. Finally, using the best setup found, we demonstrate that without any supervision with true actions, promptable representations can significantly improve latent action quality and downstream performance, increasing the success rate six-fold (Section 6).

## 2 BACKGROUND

**Problem setting.** We consider a setting of offline imitation learning from observation [32, 46], which closely matches the regime increasingly utilized by the field of embodied AI [33, 7, 5] (e.g. robotics). Our goal is to pre-train a policy $\pi(o|a)$, given a large dataset of expert trajectories $\mathcal{D} := \{(o_i^n)\}_{i=1}^{\mathcal{T}}$, containing observations but not actions (e.g. videos), and a limited number of real action labels. Ideally, the pre-trained agent should achieve maximum performance (e.g. success rate) in the environment after fine-tuning with a minimum amount of action-labeled data. The commonly considered ratio of labeled to unlabeled data is around $2 - 10\%$ in the existing work [57, 36], while in our experiments, we consider a ratio as low as $< 1\%$.

**Promptable representations.** A VLM processes multimodal inputs by encoding both the image and text before generating an output. The resulting sequence of token embeddings has length I + P + O, where I represents the visual token count, P represents the prompt token count, and O corresponds to the number of tokens produced during generation. We follow the Chen et al. [9] and define promptable representations simply as a process of obtaining observation embeddings from the VLMs given a task-specific prompt and some extraction and aggregation strategy. We obtain such representations from the last and next-to-last layers [9]. In contrast to the Chen et al. [9], Huang et al. [24] we cannot learn pooling from the data to better predict true actions or obtain better reward. Thus, we experiment only with simple fixed strategies, such as taking the mean over all embeddings or taking only the embedding of the last token from either prompt or the generated answer. Thus, the final promptable representations is always just a single vector $s \in R^D$, where $D$ is a VLMs hidden size.

**Latent action models.** Given the dataset of observations $\mathcal{D} := \{(o_i^n)\}_{i=1}^{\mathcal{T}}$, latent action models (LAM) [41, 17, 42] try to infer latent actions $z_t$ such that they are maximally predictive of observed transitions $(o_t, o_{t+1})$ while being minimal [42], i.e. describe changes only relevant to control. After pre-training, LAM is used to supply latent actions for behavioral cloning (BC) on unlabeled dataset to obtain useful behavioral priors. As a final stage, small decoder is trained to map from latent to ground-truth actions on a small number of labels.

Modern LAMs [6, 53, 12, 11, 10, 18] mostly follow the same high-level architecture introduced by LAPO [42], which uses a combination of inverse (IDM) and forward (FDM) dynamics models to infer latent actions. Given a transition $(o_t, o_{t+1})$, IDM first infers latent action $z_t \sim f_{\text{IDM}}(\cdot|o_t, o_{t+1})$, which FDM further uses to predict the next observation $\hat{o}_{t+1} \sim f_{\text{FDM}}(\cdot|o_t, z_t)$. Both models are trained jointly to minimize the loss $\mathcal{L}_{\text{MSE}} = \mathbb{E}_{(o_t, o_{t+1}) \sim \mathcal{D}} \left[ \|f_{\text{FDM}}(f_{\text{IDM}}(\boldsymbol{o}_t, \boldsymbol{o}_{t+1}), \boldsymbol{o}_t) - \boldsymbol{o}_{t+1}\|^2 \right]$.

**Limitations of latent action models.** Recent studies highlighted LAM failure when action-correlated distractors are present [36, 28, 56]. While they can recover ground-truth actions when only controllable changes are present, real-world videos typically involve both controllable factors and exogenous noise (e.g., people moving in the background). In such cases, LAMs cannot disentangle the dynamics, leading latent actions to primarily capture noise, which makes them useless for imitation learning. Both Nikulin et al. [36], Zhang et al. [56] proposed providing supervision with a small number of true actions during LAM training to help identify controllable changes. While this solution is effective, it is not generalizable, as in many domains, such as egocentric human videos [52], it is not possible to obtain true actions in a reasonable way.

## 3 EXPERIMENTAL SETUP

**Environments and datasets.** In contrast to Nikulin et al. [36], we use MetaWorld Multi-Task 10 [54] as our primary benchmark, as it provides greater realism than Distracting Control [44], while being lightweight enough to allow experimentation with VLMs under limited resources. We modify

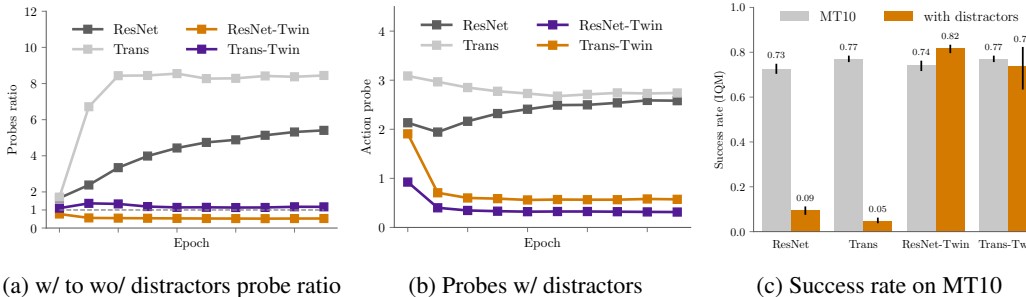

(a) w/ to wo/ distractors probe ratio    (b) Probes w/ distractors    (c) Success rate on MT10

Figure 2: Demonstration that quality of latent actions learned by LAPO completely degrades in the presence of distractors, which results in almost zero success rate. Action probe represents MSE of a linear probe trained to predict real actions from latent actions. See Section 3 for detailed explanation. We show that with the ideal target for FDM, which perfectly disentangles controllable features from the noise, performance may be restored, serving as a main motivation for us to explore promptable representations. We use three random seeds and report IQM and 95%-CI based on stratified bootstrapping, following the Agarwal et al. [1]. See Section 4 for details.

MetaWorld to include distracting dynamics videos in the background, using the same DAVIS videos as in Nikulin et al. [36]. We also move the default camera position farther back and remove borders around the table to include more of the background video in the observation, making latent action learning more challenging. See Figure 1 for a visualization.

We follow the standard three-stage pipeline [42, 53, 36]: (1) pre-train the LAM, (2) train behavioral cloning (BC) agent on latent actions, and (3) train a decoder head on a small number of true-action labels. For each task, we collect 5k trajectories from the scripted experts provided by MetaWorld and up to 16 additional labeled trajectories for the final stage, which is less than 1% of the full datasets.

**Evaluation.** For evaluation, we follow standard metrics similar to Nikulin et al. [36]: **action probe** and **success rate**. Specifically, we train linear probes to predict real actions from the latent ones during LAM training, while stopping the gradient through the latent actions. The final MSE serves as our quality metric, as it indicates whether the latent actions encode the real ones. This metric is also used for hyperparameter tuning, which may be impractical in real-world settings but allows us to estimate the upper-bound performance of each method for fair comparison.

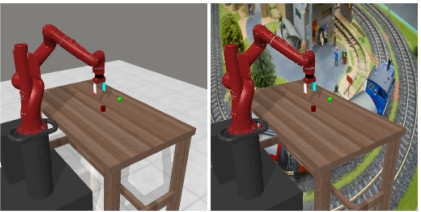

Figure 1: Visualization of observations with and without distractors in our modification of MetaWorld environment.

However, as Nikulin et al. [36] notes, linear probing has a key limitation: it can reveal whether true actions are present in the latent space, but it does not guarantee minimality, meaning that exogenous noise may still be encoded. To preserve this guarantee, we fix the latent action dimensionality to 128 for all methods, which at least allows us to rank quality under equal information bottleneck. Finally, to measure the true usefulness of latent actions, we evaluate the success rate in the environments after fine-tuning on true action labels.

**Latent action model architecture.** We use the architecture proposed by Schmidt & Jiang [42], omitting action quantization, due to its harmful effect [36, 31, 51]. We use frame stacking, but only in IDM, while FDM uses only the current frame to predict the next, as in Chen et al. [10]. Other than that, in our main experiments, we do not use any improvements upon LAPO (if not explicitly stated otherwise), such as augmentations or multi-step predictions in FDM [36, 10, 53, 12], to remove possible confounders on latent action quality. When predicting in the latent space instead of images, we follow Nikulin et al. [36] and use multiple MLP blocks similar to those used in Transformers [47]. For action decoder head, we use a small three-layer MLP. See Section C for hyperparameters used.

## 4 THE IMPORTANCE OF RIGHT TARGET

We begin with a demonstration experiment to show that the limitation of LAMs in the presence of distractors arises entirely from the poor target used in the forward dynamics model (FDM), rather than from any flaw in the overall idea or architecture. By LAM construction, latent actions are optimized

to maximally explain the dynamics. Therefore, the root of the failure to recover true actions lies in the dynamics we predict, which is directly determined by the target in FDM: $\hat{o}_{t+1} \sim f_{\text{FDM}}(\cdot|o_t, z_t)$. What would be the ideal target for FDM? And if it exists, what would be the final performance? Could LAM recover the ground-truth actions despite distractors in the input observations to IDM and FDM? If not, the idea with promptable representations would be impractical.

**Setup.** To answer these questions we construct a special dataset with twin observations for each task: during data collection we render and save same observation with and without distractors. Next, during training we feed observations with distractors as inputs to IDM and FDM, but as the target for FDM we use next observation without distractors. As the actual controllable changes are preserved (the underlying state is the true next state), it serves as a target with ideal disentanglement of controllable features from exogenous noise (see Figure 1). To show that existing limitations are agnostic to the architecture of FDM and IDM, we explore both ResNet [42] and spatio-temporal transformer [6, 53] backbones.

**Results.** First of all, as can be seen in Figure 3, we confirm that in our domain simply adding distractors results in complete degradation of latent actions quality regardless of backbone used. This subsequently leads to almost zero success rate after fine-tuning on true actions (see Figure 2c), which does not happen without distractors. Ideally, probes should be close to each other, as real underlying actions are identical between both settings.

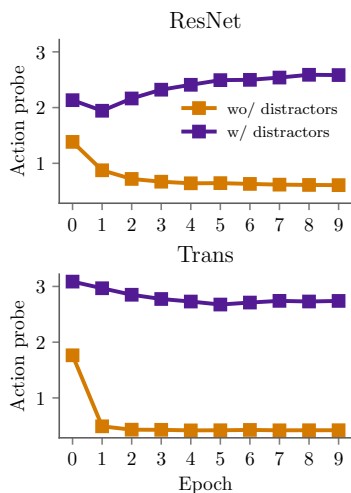

Figure 3: Baseline LAPO action probes on MT10. Averaged over 3 random seeds. Action probe represents MSE of a linear probe trained to predict real actions from latent actions. See Section 3 for detailed explanation.

Next, in Figures 2a and 2b we show the effect of using perfect targets during LAPO training (with -Twin postfix). To better illustrate the trend, in Figure 2a we report the ratio of probes with and without distractors for each method. With the ideal target probes immediately drop to the level of LAPO without distractors, and ratio converges to one. To our surprise, it is in fact possible to get even better result, as LAPO-ResNet achieves ratio below one, i.e. outperforming LAPO-ResNet without distrators. We attribute this to the implicit augmentation effect of distractors. Finally, improvement in latent action quality directly results in leveling success rates (see Figure 2c).

Overall, this result supports that the right target is the key to unlock latent action learning in the presence of distractors. Although these experiments may seem obvious in hindsight, they allow us to convey a key empirical observation about latent action learning, one that provides the same intuition that originally led us to explore promptable representations.

## 5 THE PROMISE OF PROMPTABLE REPRESENTATIONS

Our main hypothesis is that VLMs, due to their common-sense reasoning abilities, can serve as an effective unsupervised way of obtaining clean observation representations, which would disentangle controllable features from the noise. As we demonstrated in the previous section, it would be enough to unlock latent action learning in the presence of distractors.

We have no doubt that most modern VLMs would easily identify the robotic arm location in the image (like Figure 1) and describe it in detail, even in the presence of background noise. However, the ability to generate valid text does not necessarily imply that the underlying embeddings are suitable for our purposes. For a representation to serve as an effective target for LAM, it should (1) contain task-centric visual information, (2) be minimal by filtering out visual details irrelevant to the prompt, and (3) remain consistent across dynamics to mimic changes caused by real actions. Unfortunately, current VLMs are known to struggle with visual focus [40, 43] and pixel-level understanding [19, 13, 30]. Given these limitations, we begin by benchmarking a wide variety of modern VLMs to assess their viability, conducting $\sim 29$k experiments in total. As an additional baseline, we included representations from self-supervised methods such as CLIP [39] and DINOv2

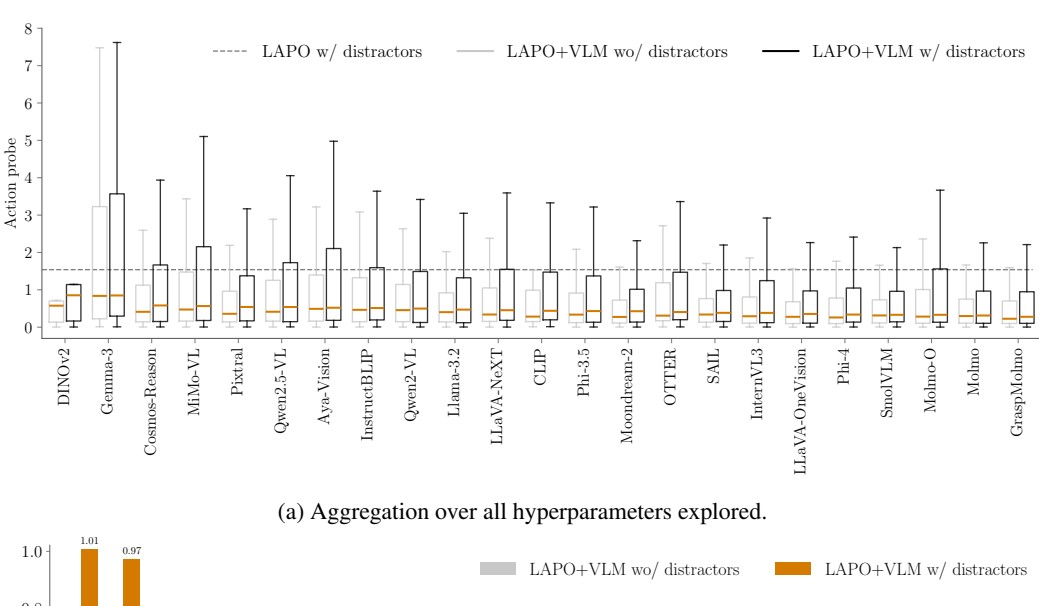

(a) Aggregation over all hyperparameters explored.

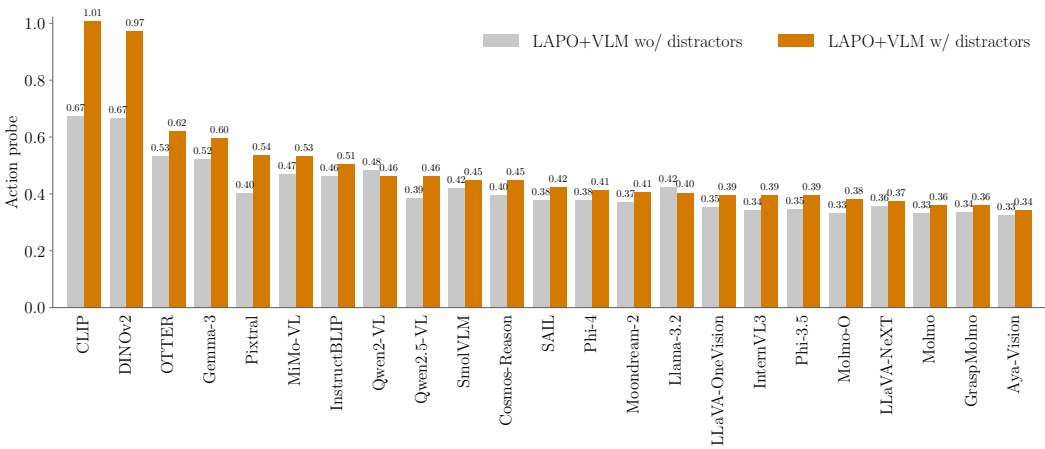

(b) Aggregation over the best hyperparameters.

Figure 4: Benchmarking the effectiveness of promptable representations provided by different VLMs for latent action learning on all tasks from MT10. Action probe represents MSE of a linear probe trained to predict real actions from latent actions. See Section 3 for detailed explanation. Results aggregated over ~ 29k experiments. Overall, all VLMs provide some improvement over LAPO, with Molmo performing the best and Gemma-3 the worst. For details and exact experimental protocol see Section 5. We additionally provide the ranking for each combination of hyperparameters in the Figure 5.

[37], which are not promptable VLMs but were pre-trained on large amounts of visual data. Based on this benchmark, we then select the most effective VLM along with the best hyperparameters (e.g., prompt, aggregation strategy, and others).

**Proper way to evaluate VLMs via small scale experiments.** Conducting large scale VLMs evaluation on the full datasets would be prohibitively expensive. Chen et al. [9] proposed assessing prompts via linear probing on small datasets, for example by asking whether task-relevant entities are present in the image and measuring probe accuracy. While feasible, this approach is suboptimal in our setting. Probing representations to predict real actions may help rank prompts for a single VLM, but it cannot reliably compare across multiple VLMs or hyperparameters, since probing does not capture the minimality of representations, an essential property for LAMs. Instead, we directly train LAPO+VLM on a small subset of trajectories, e.g. 64 instead of full 5k, and measure the resulting latent action quality. We validated that hyperparameter rankings obtained in this way transfer reasonably well to the full dataset, although probes can have different values.

**Bechmarking general VLMs.** We summarize our main benchmarking results in Figure 4 and provide full per-hyperparameter rankings in Figure 5. For each VLM, we evaluated eight prompt variants designed in different styles to exploit diverse VLM capabilities (e.g., CLIP-style captions, pointing,

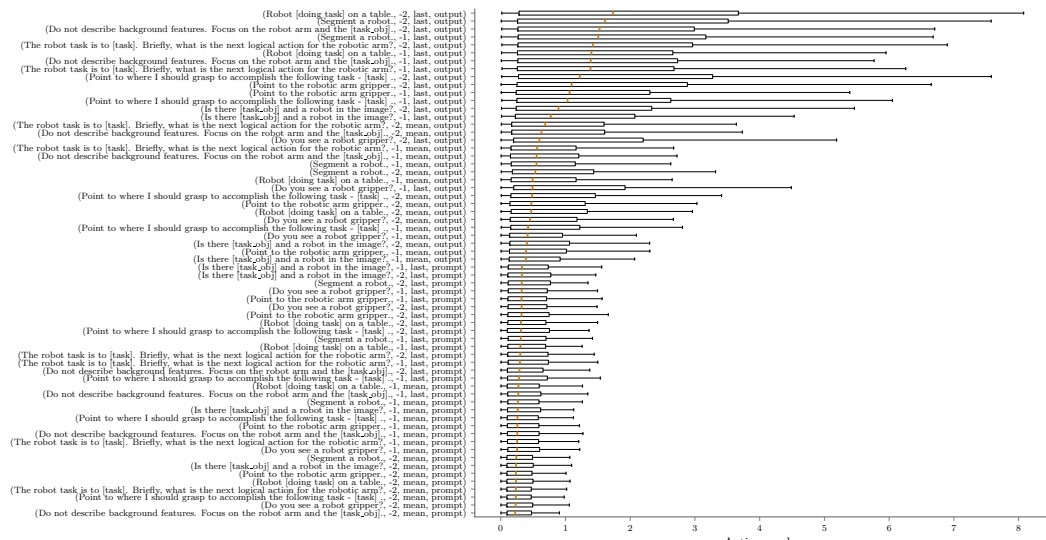

Figure 5: Action probe rankings across all explored hyperparameter combinations. Smaller probe is better. Hyperparameters in order: prompt, layer, reduction type and source of the embeddings. Reported values are averaged over all VLMs, tasks, and settings (with and without distractors). Feel free to zoom in!

segmentation; see Table 1 in Section B). We further varied the source of representations (last vs. next-to-last layer, prompt vs. generated embeddings) and the aggregation strategy (averaging vs. last non-padding token). This yields 64 runs per VLM, per task, per dataset, amounting to roughly 29k experiments in total (including VLMs which we will explore later). The full list of VLMs, including exact model names, sizes, and prompt templates, is provided in Section B.

As can be seen in Figure 4a, overall all VLMs provide some degree of improvement over LAPO in terms of the median action probe. However, some of them, especially Molmo [16], are generally preferable and have lower variance, indicating higher robustness to different hyperparameters. In Figure 4b we visualize ranking by averaging best scores for each task. While this changes ranking a bit, we still observe that Gemma-3 [45] is the worst and Molmo [16] is consistently the best. Based on Figure 5, we observe that in general, promptable representations aggregated by averaging next-to-last layer prompt embeddings perform the best. From a practical standpoint, this is beneficial, as it eliminates the additional time spent on answer generation. Ironically, the best prompt is *Do not describe background features. Focus on the robot arm and the [task-obj]*, which explicitly asks VLM to filter out distractors.

This brings us to a striking conclusion that state-of-the-art VLMs do not necessarily provide better promptable representations. For example, InstructBLIP [14] outperforms both Gemma-3 [45] and Pixtral [2], despite being considerably older. Furthermore, Cosmos-Reason [4] results indicate that explicit fine-tuning on robotics data is not sufficient to guarantee improved representations. We believe that our results, besides relevance to LAMs, reveal a large blind spot in how VLMs are currently evaluated, with critical implications for robotics and VLA models.

On the other hand, the results from DINOv2 and CLIP (see Figure 4) highlight the vital importance of language conditioning. Although DINOv2 and CLIP may possess helpful inductive biases, for example by attending to moving objects, without language conditioning there is no guarantee that these objects are the ones that are controllable. Both methods achieve the worst latent action quality among all approaches we considered. For instance, OTTER uses the same CLIP model but applies simple training-free filtering using text CLIP embeddings. This small modification significantly improves latent action quality, although still not to the level of native VLMs such as Molmo.

**Benchmarking embedding VLMs.** In our main benchmark (see Figure 4), we evaluated conventional VLMs, which were not explicitly trained to produce strong unified representations and therefore required heuristics such as embedding averaging. Recently, a new class of embedding VLMs has emerged [27, 34]. These models are designed specifically to learn high-quality, promptable, and multimodal embeddings for zero-shot retrieval. Given the similarity of their objective to ours, one might expect them to perform better. To test this, we evaluated three recent state-of-the-art models

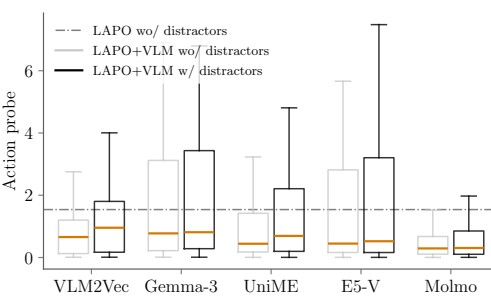 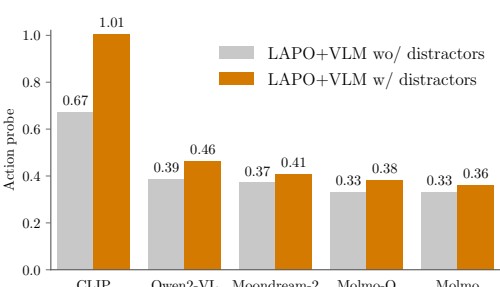

Figure 6: Benchmarking the effectiveness of promptable representations provided by recent *embedding* VLMs for latent action learning on all tasks from MT10. Overall, embedding VLMs, despite their promise, do not deliver any substantial gains compared to traditional VLMs, such as Molmo. We include Gemma-3 and Molmo results here for convineince.

Figure 7: Molmo performance investigation, using aggregation over the best hyperparameters. We benchmark both Molmo versions, which both use CLIP but differ in LLM backbones (OLMo vs Qwen2), as well as Qwen2-VL. Since both Molmo share the pre-training data but differ in architecture, we conclude that the likely source of their superior performance lies in the data.

[34, 23, 26] using the same protocol as earlier, but separately as they require different prompt formats. As can be seen in Figure 6, such models do not deliver any substantial gains. In fact, VLM2Vec-V2 [34], best model in its class, performed worse than Gemma-3, which was the weakest model in the main benchmark, and none of the models surpassed Molmo. Our results indicate that embedding VLMs do not actually encode only prompt-specific visual information into the embeddings and fail to deliver the anticipated benefits.

**Why does Molmo perform so well?** Given Molmo's strong performance, it is natural to ask what drives its improved representations. Directly answering this is difficult, but we can gather indirect evidence suggesting that the gains stem primarily from pre-training data rather than from the specific LLM or vision encoder architecture. Fortunately, Molmo provides two variants: Molmo-D, which uses Qwen2 as its backbone [50], and Molmo-O, which uses OLMo [22], while both employ CLIP [39] as the vision encoder. In contrast, Qwen2-VL [49] does not use CLIP, offering a useful comparison point to disentangle architectural effects. We therefore benchmarked and compared these models, as shown in Figure 7. The results show that CLIP alone performs the worst, Molmo-O ranks second after Molmo-D, and Qwen2-VL performs worse still. Since the Molmo variants share the same pre-training data but differ in backbone architecture, we conclude that the likely source of their superior performance lies in the data rather than the architecture. A further hypothesis is that Molmo's advantage may come from its visual pointing abilities, but this seems unlikely since Moondream-2 also has this ability yet performs worse.

## 6 PROMPTABLE REPRESENTATIONS UNLOCK TASK-CENTRIC LATENT ACTIONS

Based on the benchmark results (see Figure 4), we selected multiple VLMs from worst to best for further experiments: Gemma-3, Phi-4, Molmo and GraspMolmo. As an additional baseline we evaluated OTTER [24] approach to promptable representations. Although all of them achieved improvements in latent action quality upon LAPO on small datasets, it remains necessary to validate whether this performance transfers to the full 5k datasets and yields improved success rates, as this is not guaranteed [36]. We chose the best hyperparameters for each VLM and trained LAPO+VLM on the full datasets, using three random seeds. As specified in Section 3 we used 16 labeled trajectories with ground-truth actions for final fine-tuning. See Section C for complete hyperparameters.

**Results.** We present the resulting action probes for each task in Figure 8 and final success rates after fine-tuning on 16 trajectories with real actions in Figure 9. As can be seen in Figure 8, LAPO+VLMs achieve a substantial improvement in latent action quality, both with and without distractors. With distractors, they nearly close the gap to LAPO trained without distractors, and without distractors, they slightly outperform it (e.g., Molmo). Note that we used the best hyperparameters, which can be

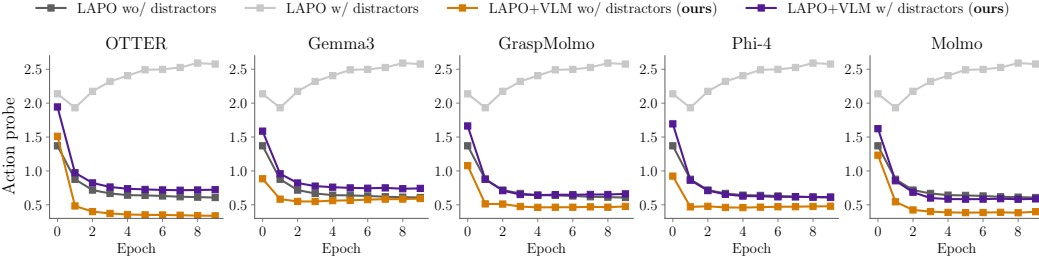

Figure 8: Action probes comparison for LAPO and LAPO+VLMs on full datasets for all tasks in MT10. Results are averaged over three random seeds. As can be seen, LAPO+VLMs significantly improves upon LAPO in terms of the latent actions quality, and without any supervision with true actions closes the gap with LAPO without distractors. While all VLMs with bring improvements, Molmo achieve best results overall, especially given it high robustness to hyperparameter choices (see Figure 4a). For resulting success rates see Figure 9. We provide per-environment probes in Figure 13, Section A.

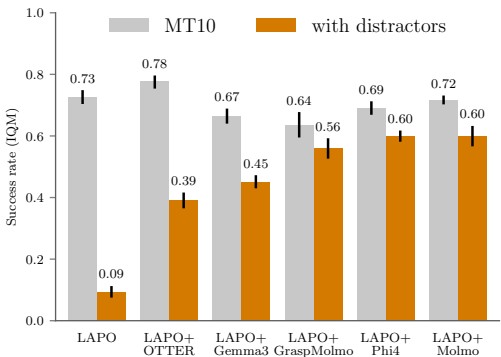

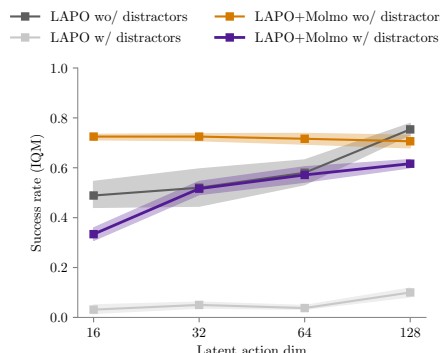

Figure 9: Success rate on MT10 for LAPO and LAPO+VLMs, which uses promptable representations. We use three random seeds and report IQM and 95%-CI based on stratified bootstrapping, following the Agarwal et al. [1].

Figure 10: Success rate on MT10 for LAPO and LAPO+Molmo, with varying latent action dimension to control the information bottleneck. We use three random seeds and report IQM and 95%-CI based on stratified bootstrapping, following the Agarwal et al. [1].

hard to find without ground-truth actions in real-world scenarios. Thus, the high robustness of Molmo to different hyperparameter choices (see Figure 4) is an important property for practical scenarios.

Crucially, the improvement in action probes on full datasets carries over to downstream performance (see Figure 9): success rates increase by a factor of six at max in the presence of distractors, while remaining almost unchanged without them. Interestingly, we found Phi-4 to outperform GraspMolmo, despite having worse probes on small datasets. On full datasets (Figure 8), however, Phi-4 is better. This indicates, that while results from a small dataset may carry over to a larger one with some error, probes on the full dataset predict the final success rate with high accuracy. We also observe that OTTHER's [24] training-free promptable representation extraction from CLIP [39] performs worse than native VLMs such as Gemma3 or Molmo. Overall our results confirm the viability of promptable representations as a clean target for latent action modeling under distracting conditions.

**Varying the latent action information bottleneck.** In all previous experiments, we fixed the latent action dimension to 128 to ensure the same level of latent action minimality across all methods. However, 128 was inherited from LAPO and may not be optimal. Therefore, we conducted additional experiments varying the latent action dimension, using full datasets and three random seeds. The results are summarized in Figure 10. We observe that promptable representations not only increase success rates in the presence of distractors but also significantly improve upon LAPO without distractors under stronger minimality constraints (e.g. 16 action dimensions). This further supports the claim that VLMs help filter out information that is not relevant to controllable changes, allowing for more compact latent action space.

## 7 DISCUSSION AND LIMITATIONS

**Segmentation, while simple, is not enough.** The concept of extracting VLM embeddings with the hope that they will filter out distractors may initially seem strange. If the goal is to filter out distractors, would not it be more straightforward to simply segment the relevant parts and train LAPO directly in image space using masks? In fact, our benchmark includes VLMs capable of segmentation, such as Sa2VA [55], and we even utilize such prompts (see Table 1), yet we still rely on embeddings instead of masks. While segmentation is appealing, it does not address the fundamental problem. Consider a scenario with a robotic arm and varying lighting conditions. Even if we segment the arm, we will still get changes in our observations that are not related to the actual actions, such as color shifts and reflections on the arm. The same issue arises with camera movement and changes in perspective. The key, therefore, is to work in a semantic latent space, which is where the common-sense reasoning capabilities of VLMs become crucial.

**On the choice of MetaWorld benchmark.** One notable limitation of our study is its small scale, as we rely on MetaWorld as our primary benchmark and do not extend our analysis to large VLAs and datasets, such as Open-X [38]. However, this choice is deliberate for two reasons. First, while MetaWorld is simple, with distractors, it is difficult enough to completely break traditional LAMs and to distinguish different VLMs in terms of the promptable representations' quality (as we demonstrate in Section 5). As an early exploration, it was crucial to expand in variety (e.g., exploring more VLMs) within our limited resources. We hope that our analysis provides practitioners with valuable insights into the available options. Second, encoding entire datasets is both expensive and time-consuming, as it involves inference with large VLMs (e.g., 8B parameters) and generating answers. For our 5k trajectory datasets, the process can quickly exceed 24 hours, let alone for truly large datasets. Since this is purely inference and gradients are not required, the process can be significantly accelerated, for example, using vLLM [29]. However, we have left this as future work.

## 8 RELATED WORK

**Latent action learning.** Imitating policy given only observations is the problem that latent action learning tries to tackle. Edwards et al. [17] suggested extracting latent actions from consecutive states with the help of some amount of true actions present. LAPO [42] scales up the approach by introducing a bottleneck between forward and backward dynamics. Building on LAPO, other approaches emerged that continued to scale latent action extraction for pre-training action models [5, 7, 58, 6, 25]. However, most of the above methods imply the presence of either noise-free datasets or an abundance of ground-truth action labels, which in general is not true for in-the-wild video data Grauman et al. [20; 21].

Some of the previous works Nikulin et al. [36], Zhang et al. [56] show that, with noise, the quality of latent actions degrades promptly, and the only proposed remedy was to increase the number of action labels. In our work, we propose a way to extract latent actions that is robust to exogenous noise and, at the same time, does not require true action labels.

**Promptable representations.** In contrast to state augmentation techniques, code generation, or reward modeling [3, 48, 15], the approach of promptable representations uses the internal embeddings of large models for performing a downstream task. Chen et al. [9] use VLM embeddings generated with a task-specific prompt to extract better state representations. Using them as input, it enhances the performance of an RL model both in Minecraft and Habitat environments. Similar work by Huang et al. [24] also employs semantic extraction by using a dot product of text and visual features from CLIP, which allows for claiming superior performance of an action model on robotic benchmarks. The important difference must be emphasized: both aforementioned works use VLM to *enhance* the performance of downstream algorithms. In contrast, when exogenous noise is present in the data, the quality of latent actions is exponentially worse [35] (than without the noise). Thus, filtering the noise with the common-sense abilities of VLMs is a way to make Latent Action Models show reasonable performance.

There exists another approach, UniVLA [8], aimed at task-specific latent action filtering. UniVLA adds the language task instruction embedding to the IDM and FDM inputs, which may help disentangle task-relevant videos from noise at a high level, but does not provide the per-step learning signal needed to accurately recover low-level ground-truth actions. As a result, during single-task learning

the task instruction is a constant vector, and UniVLA effectively reduces to LAPO. This is a general limitation of UniVLA, whereas we show that promptable representations remain effective even in the single-task learning regime.

# 9 CONCLUSION

In this work, we demonstrated that promptable representations provided by Vision-Language Models can effectively filter out action-correlated distractors, enabling task-centric latent actions. Our experiments on the Distracting MetaWorld benchmark confirmed that using task-centric promptable representations as targets for LAPO substantially improves both latent action quality and downstream success rates. We hope that our results will inspire the community to explore promptable representations at scale for the next generation of Vision-Language-Action models.

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

# A  ADDITIONAL FIGURES

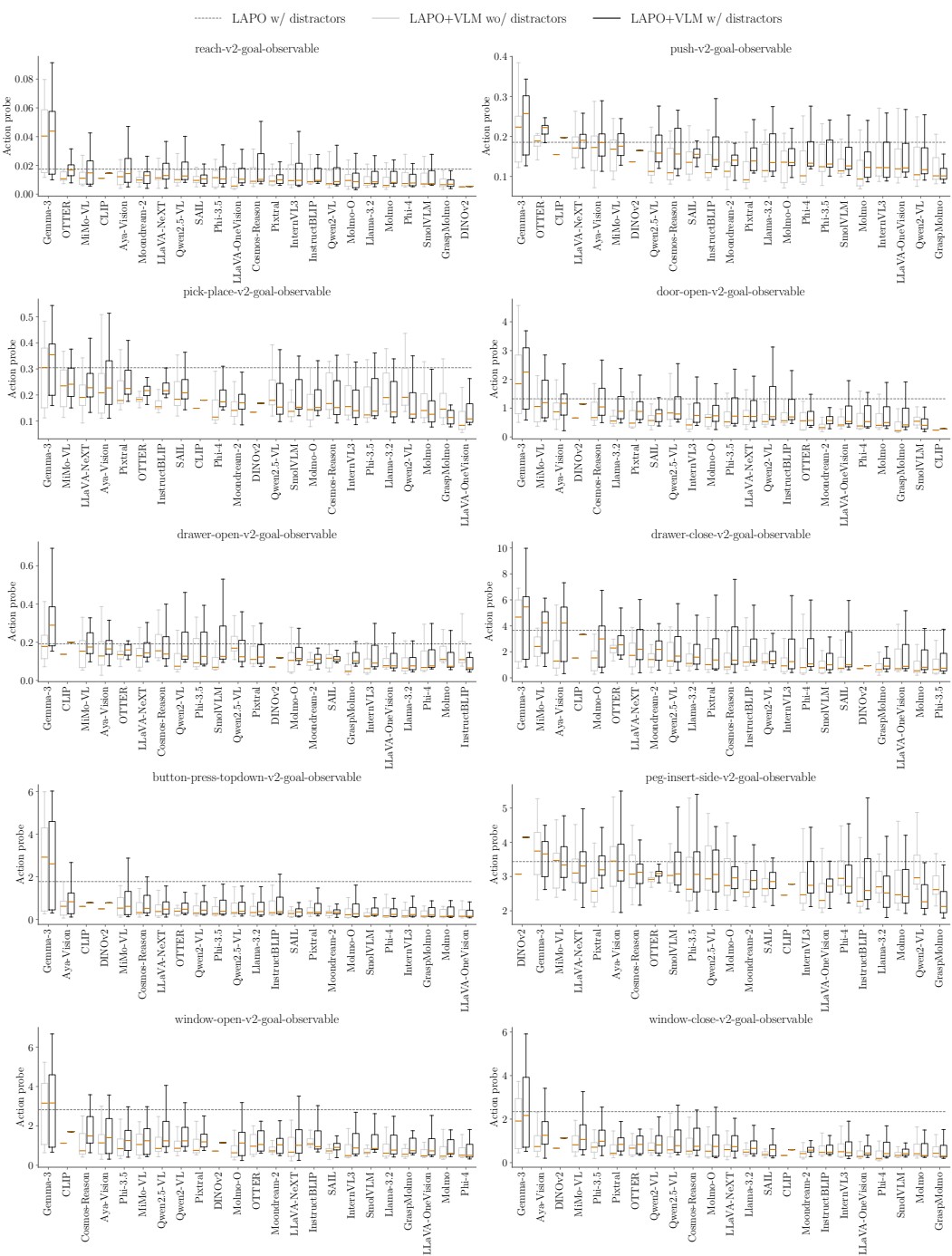

Figure 11: Aggregation over all hyperparameters for each task in MT10.

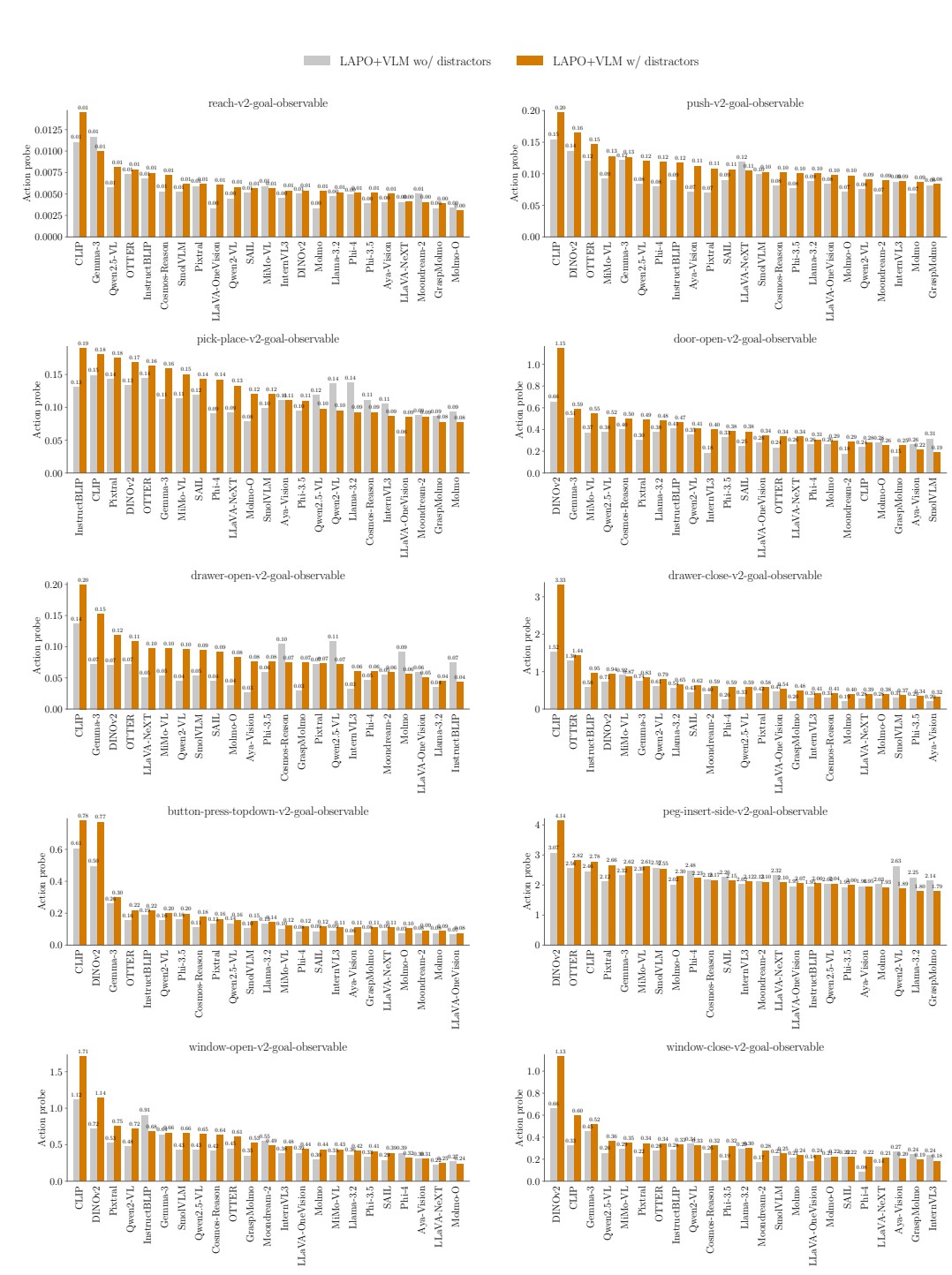

Figure 12: Probe values for best hyperparameters for each task in MT10.

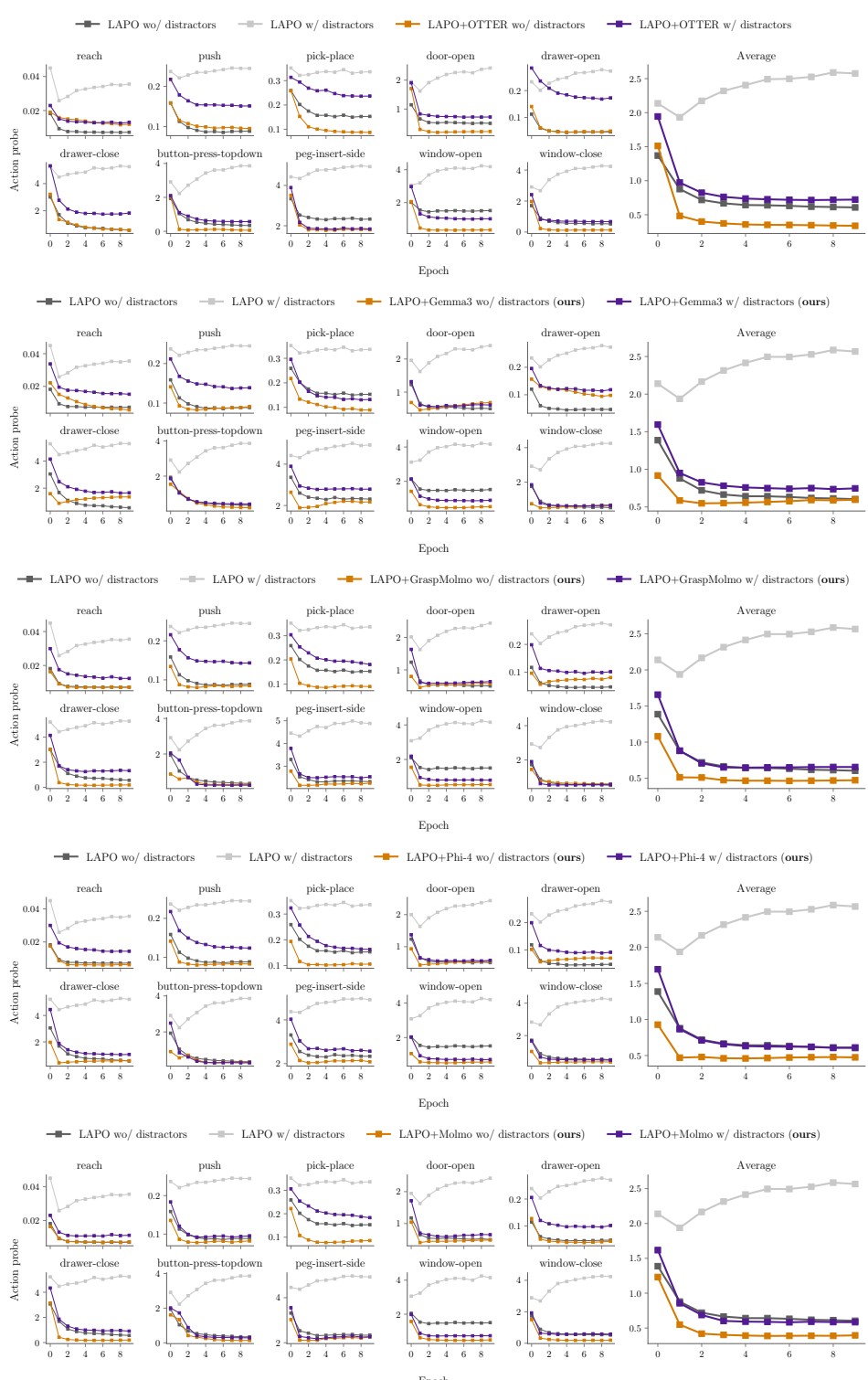

Figure 13: Action probes comparison for LAPO and LAPO+VLMs on full datasets for all tasks in MT10. Results are averaged over three random seeds. As can be seen, LAPO+VLM significantly improves upon LAPO in terms of the latent actions quality, and without any supervision with true actions closes the gap with LAPO without distractors. While all VLMs bring improvements, Molmo achieve best results overall. For resulting success rates see Figure 9

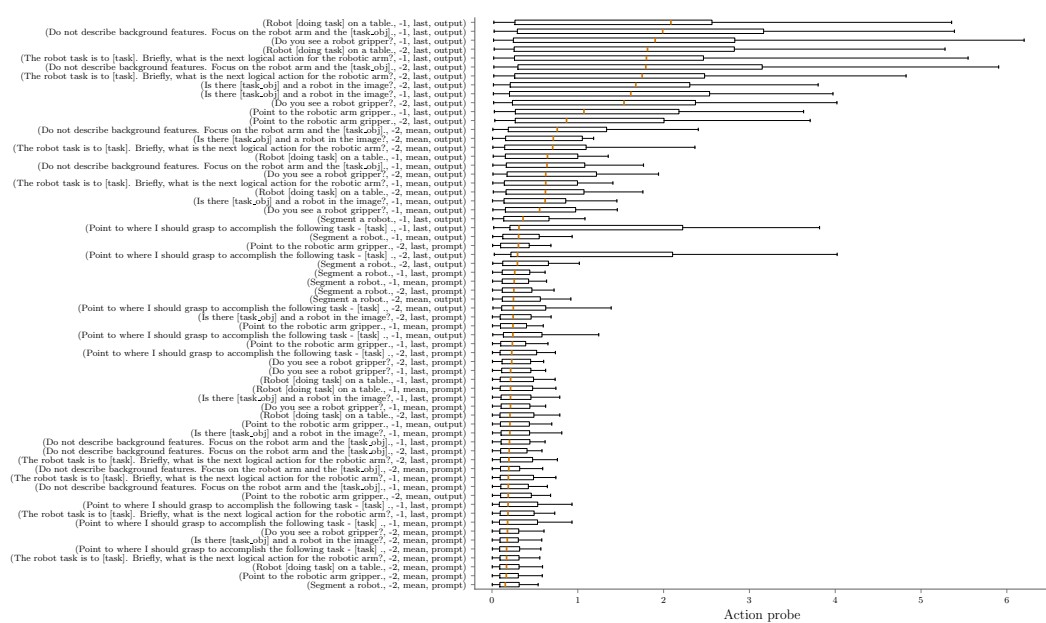

Figure 14: Action probes ranking for all combinations of hyperparameters explored for Molmo VLM. Values are averaged over all tasks and settings, e.g. with and without distractors.

# B  VISION-LANGUAGE MODELS DETAILS

Table 1: Prompt templates used in our experiments. We adapt them to each task by inserting information relevant to the task. All VLMs explored share the same prompts per task.

| Prompt |
| --- |
| The robot task is to [task]. Briefly, what is the next logical action for the robotic arm? |
| Do not describe background features. Focus on the robot arm and the [task-obj]. |
| Do you see a robot gripper? |
| Is there [task-obj] and a robot in the image? |
| Robot [doing task] on a table. |
| Point to the robotic arm gripper. |
| Point to where I should grasp to accomplish the following task - [task]. |
| Segment a robot. |

Table 2: Exact HuggingFace IDs for all VLMs we used. We shortened their names in Figures to save some space.

| Name | HuggingFace ID |
| --- | --- |
| InstructBLIP | Salesforce/instructblip-vicuna-7b |
| Molmo | allenai/Molmo-7B-D-0924 |
| Gemma-3 | google/gemma-3-12b-it |
| Llama-3.2 | unsloth/Llama-3.2-11B-Vision-Instruct |
| Qwen2.5-VL | Qwen/Qwen2.5-VL-7B-Instruct |
| InternVL3 | OpenGVLab/InternVL3-8B |
| Cosmos-Reason | nvidia/Cosmos-Reason1-7B |
| Phi-4 | microsoft/Phi-4-multimodal-instruct |
| LLaVA-OneVision | llava-hf/llava-onevision-qwen2-7b-ov-hf |
| SmolVLM | HuggingFaceTB/SmolVLM2-2.2B-Instruct |
| Pixtral | mistral-community/pixtral-12b |

## C HYPERPARAMETERS

Table 3: LAPO-ResNet hyperparameters. Names are exactly follow the configuration files used in code.

| Part | Parameter | Value |
|---|---|---|
| General | frame_stack | 4 |
| | probe_learning_rate | 0.0003 |
| | disable_distractors | True |
| | seed | 0 |
| | eval_seed | 0 |
| | eval_episodes | 50 |
| Latent action learning | latent_action_dim | 128 |
| | idm_encoder_scale | 5 |
| | idm_encoder_num_res_blocks | 1 |
| | idm_encoder_channels | [16, 16, 32, 32, 128, 128, 256] |
| | fdm_encoder_scale | 1 |
| | fdm_encoder_num_res_blocks | 1 |
| | fdm_encoder_channels | [16, 16, 32, 32, 128, 128, 256] |
| | num_epochs | 10 |
| | batch_size | 64 |
| | learning_rate | 0.0001 |
| | weight_decay | 0.0 |
| | warmup_epochs | 1 |
| | grad_norm | - |
| Latent behavior cloning | num_epochs | 10 |
| | batch_size | 64 |
| | learning_rate | 0.0001 |
| | weight_decay | 0.0 |
| | warmup_epochs | 0 |
| | encoder_scale | 5 |
| | encoder_num_res_blocks | 1 |
| | encoder_channels | [16, 16, 32, 32, 128, 128, 256] |
| Latent actions decoding | total_updates | 100000 |
| | batch_size | 64 |
| | learning_rate | 0.001 |
| | hidden_dim | 128 |
| | num_labeled_trajectories | [16, 8, 2, 4] |

Table 4: LAPO-Trans hyperparameters. Names exactly follow the configuration files used in code.

| Part | Parameter | Value |
|------|-----------|-------|
| General | frame_stack | 4 |
| | probe_learning_rate | 0.0003 |
| | disable_distractors | True |
| | seed | 0 |
| | eval_seed | 0 |
| | eval_episodes | 50 |
| Latent action learning | latent_action_dim | 128 |
| | patch_size | 32 |
| | fdm_use_cross_attn | False |
| | idm_hidden_dim | 896 |
| | idm_num_layers | 4 |
| | idm_num_heads | 16 |
| | fdm_hidden_dim | 256 |
| | fdm_num_layers | 4 |
| | fdm_num_heads | 8 |
| | normalize_qk | False |
| | pre_norm | True |
| | num_epochs | 10 |
| | batch_size | 64 |
| | learning_rate | 0.0001 |
| | weight_decay | 0.0 |
| | warmup_epochs | 1 |
| | grad_norm | - |
| Latent behavior cloning | num_epochs | 10 |
| | batch_size | 64 |
| | learning_rate | 0.0001 |
| | weight_decay | 0.0 |
| | warmup_epochs | 0 |
| | encoder_scale | 5 |
| | encoder_num_res_blocks | 1 |
| | encoder_channels | [16, 16, 32, 32, 128, 128, 256] |
| Latent actions decoding | total_updates | 100000 |
| | batch_size | 64 |
| | learning_rate | 0.001 |
| | hidden_dim | 128 |
| | num_labeled_trajectories | [16, 8, 2, 4] |

Table 5: LAPO+VLM hyperparameters. Names exactly follow the configuration files used in code.

| Part | Parameter | Value |
|---|---|---|
| General | frame_stack | 4 |
| | probe_learning_rate | 0.0003 |
| | disable_distractors | True |
| | seed | 0 |
| | eval_seed | 0 |
| | eval_episodes | 50 |
| VLM (example) | type | molmo |
| | prompt | Point to the robotic arm gripper. |
| | layer | 27 |
| | target | output |
| | reduce_strategy | mean |
| Latent action learning | latent_action_dim | 128 |
| | idm_encoder_scale | 5 |
| | idm_encoder_num_res_blocks | 1 |
| | idm_encoder_channels | [16, 16, 32, 32, 128, 128, 256] |
| | fdm_hidden_dim | 1024 |
| | fdm_num_layers | 4 |
| | fdm_expand | 4 |
| | num_epochs | 200 |
| | batch_size | 64 |
| | learning_rate | 0.0001 |
| | weight_decay | 0.0 |
| | warmup_epochs | 1 |
| | grad_norm | - |
| Latent behavior cloning | num_epochs | 10 |
| | batch_size | 64 |
| | learning_rate | 0.0001 |
| | weight_decay | 0.0 |
| | warmup_epochs | 0 |
| | encoder_scale | 5 |
| | encoder_num_res_blocks | 1 |
| | encoder_channels | [16, 16, 32, 32, 128, 128, 256] |
| Latent actions decoding | total_updates | 100000 |
| | batch_size | 64 |
| | learning_rate | 0.001 |
| | hidden_dim | 128 |
| | num_labeled_trajectories | [16, 8, 2, 4] |

