# OpenReview forum: "Vision-Language Models Unlock Task-Centric Latent Actions"
_ICLR.cc/2026/Conference — Submitted to ICLR 2026_

### Official Review · Reviewer_mdb4 · 2025-10-30

**Soundness:** 3
**Presentation:** 2
**Contribution:** 2
**Rating:** 4
**Confidence:** 3

**Summary:**

This paper describes how vision-language models (VLM) representations can be used to train latent-action models (LAM) that are robust to “action-correlated distractors”. Extensive experiments are conducted on a modified version of the MetaWorld benchmark, where the background is artificially modified to include distractors, called Distracting MetaWorld. First, the weakness of current LAM on the task is demonstrated. Then, a study on the best VLMs for the task is conducted. Finally, the effectiveness of the approach is demonstrated by studying success rate on Distracting MetaWorld.

**Strengths:**

- The robustness of latent-action model to action-correlated perturbation is a fundamental problem that differentiates humans to current systems and that should be explored by the community. This paper attempts to find a minimal setup with artificial perturbations, which is a valid and interesting approach.

- The experimental results provide a clear signal on the proposed task that VLM representations help overcome the challenge of action-correlated perturbations.

- The paper clearly acknowledges several limitations such as the simplicity of the constructed task: Perturbed MetaWorld, and the use of VLM representation as opposed to a semantic segmentation approach.

**Weaknesses:**

- The clarity of the paper could be improved. First, the notion of promptable representation is not properly introduced, or at least way too far in the paper, in section 3, L123, which might confuse readers not aware of the literature. I would suggest putting a proper definition earlier in the paper. Then, the experimental setup is not formalized properly, the author describes inputs and outputs of their IDM and FDM in section 4 without describing the specific architecture they use. It might be useful to include a Figure summarizing the pipeline. There are other presentation issues: line 107: BC is not defined, finally in Figure 2, 3 a) b), 4), it is very unclear what the y-axis represents; both for people only looking at the figure and reading the captions, and for people reading the text, paragraph “results”, line 183, should be where the metric is explained.

- Although the problem of action-correlated distractors is correctly identified, the specific construction of synthetic distractors proposed by the paper is not fully-convincing and feels a bit too artificial to be convinced that it would transfer to real world problems. Only the background is altered, with completely random and unrealistic patterns. More interesting distractors would have been: occlusion of the action, other robots in the same scenes, humans performing actions in the scene, or lightning parameters such as luminosity.

- I have a fundamental criticism about VLM representations. It is well known that representations from generative models such as LLM/VLM are not very good and it seems odd to push to use them. The paper mentions these representations should “be minimal by filtering out visual details irrelevant to the prompt”. This is exactly what self-supervised learning representations and in particular representation from joint-embedding predictive architectures (JEPA) is trying to achieve, and it should be much better suited compared to VLM representations that have the disadvantages, as shown in the paper, to be sensitive and to rely on good prompts, as shown in the paper: “Ironically, the best prompt is “Do not describe background features. Focus on the robot arm and the [task-obj]”. SSL representations also do not have the disadvantages of segmentation as described in the limitation section.

- It is not clear in this work or similar work on LAM, what is the importance of the labeled trajectories. What if we do not fine-tune on these trajectories ? What if we only train on these trajectories ?

- In conclusion, the paper reasonably answers its own hypothesis, but the setting chosen and specific distractor + the choice of VLMs representation do not convince me that this is the best way to train LAMs.

**Questions:**

- Paragraph “Why does Molmo perform so well?” explains why the data might be the primary factor that makes good promptable representation, what do you think is the right data to train a VLM in order to get strong promptable representations for LAMs ?

- In line 257 mentions the use of “64 instead of full 5k” trajectories. How reliable is it to use so few trajectories ? Later the paper mentions “we found Phi-4 to outperform GraspMolmo, despite having worse probes on small datasets”, is that the result of validating hyper-parameters on so few trajectories ?

---

> ### Author Response · Authors · 2025-11-28
> **Official Comment by Authors (1/2)**
>
> We thank the reviewer for their feedback and for carefully engaging with our work. We apologise for the delayed response; we experienced issues accessing computing resources.
>
> We now answer the following concerns raised in the review.
>
> > First, the notion of promptable representation is not properly introduced, or at least way too far in the paper, in section 3, L123, which might confuse readers not aware of the literature. I would suggest putting a proper definition earlier in the paper.
> >
>
> You are right, we moved the explanation to the Background section (Section 2) and expanded it slightly (we highlighted all text updates with the blue color).
>
> > Then, the experimental setup is not formalized properly, the author describes inputs and outputs of their IDM and FDM in section 4 without describing the specific architecture they use.
> >
>
> We apologise if this was unclear. We describe the architecture used in the Experimental Setup section (Section 3), in the paragraph beginning with “latent action model architecture” (lines 132-133). We mostly follow LAPO and LAOM exactly, without changing the architectures in any substantial way. The experimental setup and evaluation is described in the same Section. We describe the environments, datasets and evaluation metrics in detail. All experiments follow this hereafter, so we decided to explain it in a single place and not repeat further.
>
> > It might be useful to include a Figure summarizing the pipeline.
> >
>
> This is an excellent suggestion, and we are actively working on it. We hope this does not critically hinder the reviewers’ understanding of the paper’s contributions. We aim to include the visualization by the end of the rebuttal or in the camera-ready version.
>
> > There are other presentation issues: line 107: BC is not defined
> >
>
> Thank you. We changed BC to explicit “behavioral cloning (BC) agent”.
>
> > Figure 2, 3 a) b), 4), it is very unclear what the y-axis represents; both for people only looking at the figure and reading the captions, and for people reading the text, paragraph “results”, line 183, should be where the metric is explained.
> >
>
> Thanks. We currently explain this metric in detail in the Experimental Setup section (Section 3), but added additional short explanation to the Figure captions with a references to the full descriptions.
>
> > Although the problem of action-correlated distractors is correctly identified, the specific construction of synthetic distractors proposed by the paper is not fully-convincing and feels a bit too artificial to be convinced that it would transfer to real world problems. Only the background is altered, with completely random and unrealistic patterns. More interesting distractors would have been: occlusion of the action, other robots in the same scenes, humans performing actions in the scene, or lightning parameters such as luminosity.
> >
>
> We agree that the environments we have chosen are fairly simple. However, as we write in the Discussion Section, despite its simplicity, MetaWorld with distractors is challenging enough to break traditional LAMs and to clearly differentiate VLMs based on the quality of their promptable representations (as shown in Figure 4). Given limited resources, we prioritized breadth over scaling up, by evaluating more VLMs. We should also point out that most of the videos in DAVIS feature humans performing actions, so we wouldn't call them completely random and unrealistic.
>
> However, we agree that adding additional distractors would be useful, so we also added a change in lighting throughout the episode (see video examples in the updated supplementary materials), in addition to the videos. Due to time constraints, we chose one task (button-press-topdown), but for the camera-ready version, we will add up the results for the entire MT10. We used 3 random seeds and full datasets with 5k trajectories. We summarize results in a table below:
>
> | Method | Success rate |
> | --- | --- |
> | LAPO wo/ distractors | 0.99 |
> | LAPO+Molmo wo/ distractors | 0.96 |
> | LAPO w/ distractors | 0.21 |
> | LAPO+Molmo w/ distractors | 0.85 |
>
> Promptable representations still deliver a significant improvement in success rate in the presence of distractors.

---

> ### Author Response · Authors · 2025-11-28
> **Official Comment by Authors (2/2)**
>
> > I have a fundamental criticism about VLM representations. It is well known that representations from generative models such as LLM/VLM are not very good and it seems odd to push to use them. The paper mentions these representations should “be minimal by filtering out visual details irrelevant to the prompt”. This is exactly what self-supervised learning representations and in particular representation from joint-embedding predictive architectures (JEPA) is trying to achieve, and it should be much better suited compared to VLM representations that have the disadvantages, as shown in the paper, to be sensitive and to rely on good prompts, as shown in the paper: “Ironically, the best prompt is “Do not describe background features. Focus on the robot arm and the [task-obj]”. SSL representations also do not have the disadvantages of segmentation as described in the limitation section.
> >
>
> Thank you for this valuable concern. As Zhang C. et al (2025) and Misra D. et al (2024) theoretically show, the limitations of LAMs in the presence of distractors are general and arise from the properties of the data, not the specific architecture or the networks scale. To successfully recover ground-truth actions, it is extremely important that dynamics, which FDM tries to model with the help of IDM, should contain only controllable and task-centric changes (see Section 4).
>
> To demonstrate that SSL representations are ill-suited for LAMs training we provide additional results with representations obtained from DINOv2 and CLIP (see updated Figure 4 and line 312). While DINOv2 may have some inductive biases, e.g. by focusing it’s attention on moving objects, without language-conditioning there is no guarantee that this objects are exactly what is controllable in the task. We benchmarked the DINOv2 representations, and showed that it achieves worst latent action quality among all considered methods.
>
> To demonstrate this even more clearly, we compared the representations from CLIP and OTTER (see updated Figure 4) in a similar setup. OTTER uses the same CLIP model, but adds simple training-free filtering on top using text CLIP embeddings. This small change significantly improves latent action quality, but not to the level of native VLMs, such as Molmo.
>
> Similar reasoning applies to representations from JEPA, as to our knowledge, they are not language-conditioned and thus can not provide task-centric representations.
>
> > It is not clear in this work or similar work on LAM, what is the importance of the labeled trajectories. What if we do not fine-tune on these trajectories ? What if we only train on these trajectories ?
> >
>
> This is an interesting question. First of all, some amount of ground-truth actions are necessary, as BC agent trained on latent actions can not act in real environment due to the action space mismatch. Thus, it is required to learn a mapping from latent action space to ground-truth action space. Given that latent actions actually contain true actions, is should be possible with very limited amount of labels.
>
> However, simply training BC on these labelled trajectories from scratch without pre-training results in a worse performance, as has been demonstrated by LAPA [8] or LAOM [3]. The reason is simple - not enough data. Unlabelled data has a larger scale, coverage and breadth, which is beneficial for pre-training.
>
> ### Questions
>
> > Paragraph “Why does Molmo perform so well?” explains why the data might be the primary factor that makes good promptable representation, what do you think is the right data to train a VLM in order to get strong promptable representations for LAMs ?
> >
>
> Unfortunately, we can only speculate, as thorough research into VLM is extremely expensive. We can hypothesize that this question can be answered end-to-end using data attribution methods, using action probe on a small evaluation dataset as a metric. We left this for a future work.
>
> > In line 257 mentions the use of “64 instead of full 5k” trajectories. How reliable is it to use so few trajectories ? Later the paper mentions “we found Phi-4 to outperform GraspMolmo, despite having worse probes on small datasets”, is that the result of validating hyper-parameters on so few trajectories ?
> >
>
> Yes, this is the result of validating hyper-parameters on the small datasets. Although this process speeds up experiment iterations many times over, it can be slightly noisy. It is important to note that when transferring to a large dataset, the worst and best models are preserved. Moreover, the validation process itself using action probes is valid, since on a larger dataset, the order by action probes is accurately transferred to the order by the success rate.

---

> ### Author Response · Authors · 2025-11-28
> **References**
>
> References:
>
> 1. Zhang, C., Pearce, T., Zhang, P., Wang, K., Chen, X., Shen, W., ... & Bian, J. (2025). What Do Latent Action Models Actually Learn?. *arXiv preprint arXiv:2506.15691*.
> 2. Huang, H., Liu, F., Fu, L., Wu, T., Mukadam, M., Malik, J., ... & Abbeel, P. (2025). Otter: A vision-language-action model with text-aware visual feature extraction. *arXiv preprint arXiv:2503.03734*.
> 3. Nikulin, A., Zisman, I., Tarasov, D., Lyubaykin, N., Polubarov, A., Kiselev, I., & Kurenkov, V. (2025). Latent action learning requires supervision in the presence of distractors. *arXiv preprint arXiv:2502.00379*.
> 4. Lachapelle, S. (2025). On the Identifiability of Latent Action Policies. *arXiv preprint arXiv:2510.01337*.
> 5. Misra, D., Saran, A., Xie, T., Lamb, A., & Langford, J. (2024). Towards principled representation learning from videos for reinforcement learning. *arXiv preprint arXiv:2403.13765*.
> 6. Lamb, A., Islam, R., Efroni, Y., Didolkar, A., Misra, D., Foster, D., ... & Langford, J. (2022). Guaranteed discovery of control-endogenous latent states with multi-step inverse models. *arXiv preprint arXiv:2207.08229*.
> 7. Levine, A., Stone, P., & Zhang, A. (2024). Multistep inverse is not all you need. *arXiv preprint arXiv:2403.11940*.
> 8. Ye, S., Jang, J., Jeon, B., Joo, S., Yang, J., Peng, B., ... & Seo, M. (2024). Latent action pretraining from videos. *arXiv preprint arXiv:2410.11758*.

---

### Official Review · Reviewer_8hu2 · 2025-10-31

**Soundness:** 2
**Presentation:** 2
**Contribution:** 2
**Rating:** 4
**Confidence:** 4

**Summary:**

This paper explores the efficacy of promptable representations from VLMs as reconstruction targets for Latent Action Models (LAM). The author’s key idea is that the common sense reasoning abilities of VLMs can better enable LAMs to learn latent actions corresponding to the controllable changes while filtering out exogenous noise (e.g., scene details, camera motion). The authors explore representations from a wide range of VLMs and perform an extensive ablation study to validate the optimal strategy to obtain promptable representations from VLMs. Performance is evaluated over 20 recent VLMs and 29,000 experiments on the MetaWorld benchmark with distractors, both latent action quality (via linear probing) and downstream task success rates after fine-tuning are measured. The authors find that these promptable representations can effectively improve the performance of LAMs in the presence of distractors.

**Strengths:**

Leveraging the common-sense reasoning capabilities of VLMs to learn stronger latent actions centered on controllable changes is well motivated and addresses an important challenge facing current LAMs

The paper conducts an extensive empirical study to validate an optimal strategy for extracting proptable representations from VLMs. This study is conducted across a wide range of recent SOTA VLMs

The promptable representations are effective and consistently improve performance over baseline LAMs in scenes with distractors

**Weaknesses:**

Overall the novelty seems limited to the reviewer. As referenced by the authors, Chen et al. [1] proposed promptable representations, and much of the evaluation setting follows Nikulin et al. [2].

While the motivation is reasonable, the underlying reason why promptable representations lead to such improvements remains unexplored and poorly understood to the reviewer
* It is unclear why promptable representations from VLMs should be preferred to other methods like UniVLA that aim to disentangle task-specific cues and exogenous noise. The authors discuss UniVLA in Section 8, but it was unclear to the reviewer why the noted differences are relevant. Why is the proposed method preferred to UniVLA for learning task-centric latent actions?
* While it is clear that promptable representations improve performance of LAMs, it is not convincing that the improvement stems from the common sense reasoning abilities of VLMs themselves. To the reviewer it seems the improvement can stem from the larger and more diverse pretraining data that VLMs have seen compared to the VQ-VAE used in the baseline. Why would representations from a model like DINOv2 (i.e., a non-promotable model with large pre-training data) be a worse reconstruction target?

The goal of latent action learning is to unlock large scale human videos (e.g., from the web), but there are no experiments (or proof of concepts) showing that promptable representations are effective for learning latent actions from real-world videos. Datasets like EgoDex [3], which contain egocentric actions with annotated hand poses, could be helpful for this validation (as the availability of hand poses enables action probing)

Minor comments on formatting
* Line 72: should be “Latent Action Models (LAM)”
* Line 111: “linear probing” is listed as one of the evaluation metrics, but in the experiments it is reported as action probing (e.g., Figure 2 and Figure 3)
* Figure 2: bolding “ResNet” and “Trans” will prevent confusion on the x-axis label in the first plot
* Figure 5: Currently this figure is a bit difficult to read and there is a lot of repetition. In future versions readability could be improved, for example through assigning letters to specific prompts, or presenting only mean and std instead of the entire box plot

[1] William Chen, Oier Mees, Aviral Kumar, and Sergey Levine. Vision-language models provide promptable representations for reinforcement learning. Arxiv, 2024

[2] Alexander Nikulin, Ilya Zisman, Denis Tarasov, Nikita Lyubaykin, Andrei Polubarov, Igor Kiselev, Vladislav Kurenkov. Latent Action Learning Requires Supervision in the Presence of Distractors. ICML, 2025

[3] Ryan Hoque, Peide Huang, David J. Yoon, Mouli Sivapurapu, Jian Zhang. EgoDex: Learning Dexterous Manipulation from Large-Scale Egocentric Video. Arxiv, 2025

**Questions:**

(Line 118) Does fixing the latent action dimension to 128 really guarantee minimality? Regardless of the dimension of the latent action, it can still end up encoding noise rather than controllable changes

What distinguishes this work conceptually or experimentally from prior studies like Chen et al. [1] and Nikulin et al. [2]?

an the authors provide analysis or evidence clarifying why promptable representations improve latent action learning performance beyond empirical gains?

---

> ### Author Response · Authors · 2025-11-28
> **Official Comment by Authors (1/2)**
>
> We thank the reviewer for their feedback and for carefully engaging with our work. We apologise for the delayed response; we experienced issues accessing computing resources.
>
> We now answer the following concerns raised in the review.
>
> > Overall the novelty seems limited to the reviewer. As referenced by the authors, Chen et al. [1] proposed promptable representations, and much of the evaluation setting follows Nikulin et al. [2].
> >
>
> We respectfully disagree. Although the components we use are not new, this does not diminish the novelty of our work. We believe novelty should be judged not by the method alone but by the new insights and results it enables.
>
> Thus, while Chen et al. and Huang et al. also use promptable representations, their improvements are in standard RL, modest and do not transform a non-functional method into one that fully recovers performance. More importantly, their results do not guarantee the existence of zero-shot properties required for Latent Action Models, since, unlike in their setup, we cannot learn representation filtering from real actions.
>
> For LAMs, representations must (1) contain task centric visual information, (2) be minimal by filtering out visual details irrelevant to the prompt, and (3) remain consistent across dynamics to mimic changes caused by real actions. Our exhaustive benchmarking shows that VLMs differ substantially in these capabilities, which was not previously known. And this is significant: choosing the right VLM turns LAM from non-functional into a method that performs comparably to distractor-free data, effectively overcoming LAM’s core limitations.
>
> As for the evaluation setting, we believe that the use of standard benchmarks and metrics does not diminish novelty, but merely simplifies comparison and reproducibility. At least, as long as the benchmarks provide meaningful signals, which we clearly demonstrated in Section 4.
>
> > It is unclear why promptable representations from VLMs should be preferred to other methods like UniVLA that aim to disentangle task-specific cues and exogenous noise. The authors discuss UniVLA in Section 8, but it was unclear to the reviewer why the noted differences are relevant. Why is the proposed method preferred to UniVLA for learning task-centric latent actions?
> >
>
> We apologise if this was not clear from the text. UniVLA simply adds the language task instruction to the IDM and FDM as input, which may help disentangle task-relevant videos from noise at a high level, but does not provide the per-step learning signal needed to accurately recover low-level ground-truth actions. Thus, during single task learning, task instruction is always a constant vector, and UniVLA converts to just LAPO. Thus,  UniVLA is simply not applicable in our setup. This is the general limitation of UniVLA, while we show that promptable representations are applicable even in the single task learning regime and provide learning signal on each step (as VLM encodes each observation into task-centric embedding).
>
> UniVLA additionally claims that learning LAMs in DINOv2 latent space further enhances “object-centric and spatially aware properties”. We benchmarked the DINOv2 representations (see updated Figure 4), and showed that it achieves worst latent action quality among all considered methods.
>
> > While it is clear that promptable representations improve performance of LAMs, it is not convincing that the improvement stems from the common sense reasoning abilities of VLMs themselves. To the reviewer it seems the improvement can stem from the larger and more diverse pretraining data that VLMs have seen compared to the VQ-VAE used in the baseline. Why would representations from a model like DINOv2 (i.e., a non-promotable model with large pre-training data) be a worse reconstruction target?
> >
>
> As Zhang C. et al (2025) and Misra D. et al (2024) theoretically show, the limitations of LAMs in the presence of distractors are general and arise from the properties of the data, not the specific architecture or the networks scale. To successfully recover ground-truth actions, it is extremely important that dynamics, which FDM tries to model with the help of IDM, should contain only controllable and task-centric changes (see Section 4).
>
> While DINOv2 may have some inductive biases, e.g. by focusing it’s attention on moving objects, without language-conditioning there is no guarantee that this objects are exactly what is controllable. We benchmarked the DINOv2 representations (see updated Figure 4), and showed that it achieves worst latent action quality among all considered methods.
>
> To demonstrate this even more clearly, we compared the representations from CLIP and OTTER (see ****updated Figure 4) in a similar setup. OTTER uses the same CLIP model, but adds simple training-free filtering on top using text CLIP embeddings. This small change significantly improves latent action quality, but not to the level of native VLMs, such as Molmo.

---

> ### Author Response · Authors · 2025-11-28
> **Official Comment by Authors (2/2)**
>
> > The goal of latent action learning is to unlock large scale human videos (e.g., from the web), but there are no experiments (or proof of concepts) showing that promptable representations are effective for learning latent actions from real-world videos. Datasets like EgoDex [3], which contain egocentric actions with annotated hand poses, could be helpful for this validation
> >
>
> You are correct, this is precisely the direction in which we intend to develop further. However, at present, we do not have sufficient computing resources to embed such a large amount of data using VLMs. We hope that the evidence already provided is sufficient to support our claims.
>
> > Line 118) Does fixing the latent action dimension to 128 really guarantee minimality? Regardless of the dimension of the latent action, it can still end up encoding noise rather than controllable changes
> >
>
> This is a valid concern. Unfortunately, there is no direct way to measure the amount of noise in latent actions (unlike a simple probe for actual actions). The most effective way is to simply train the BC agent on the resulting actions and test it in a real environment. If the actions contain a lot of noise, they will be useless for pre-training. Since in our case we get large increases in the success rate, we can confidently say that their quality is significantly improved, which in turn leads to the conclusion that there is less noise.
>
> To provide additional comparison, we re-run the main experiments with varying latent action dimensions, from 16 up to original 128, on full datasets from all environments and with 3 random seeds. We report success rate IQM and 95% CI. Results provided below (and in updated Section 6, line 424):
>
> | Method \ Latent action dim | 16 | 32 | 64 | 128 |
> | --- | --- | --- | --- | --- |
> | LAPO wo/ distractors | 0.49 (0.44, 0.55) | 0.52 (0.45, 0.6) | 0.58 (0.53, 0.63) | 0.75 (0.73, 0.78) |
> | LAPO+Molmo wo/ distractors | **0.72** (0.71, 0.73) | **0.72** (0.71, 0.74) | **0.72** (0.7, 74) | **0.71** (0.68, 0.73) |
> | LAPO w/ distractors | 0.03 (0.02, 0.05) | 0.05 (0.04, 0.06) | 0.05 (0.03, 0.05) | 0.1 (0.08, 0.12) |
> | LAPO+Molmo w/ distractors | **0.33** (0.31, 0.36) | **0.5** (0.49, 0.55) | **0.57** (0.54, 0.6) | **0.62** (0.6, 0.63) |
>
> Interestingly, we observe that promptable representations not only allow to increase success rates in the presence of distractors, but significantly improve upon LAPO without distractors with larger minimality constraints (e.g. only 16 action dim). This further confirms that VLMs help filter out information not relevant to the controllable changes (e.g. static background).
>
> > What distinguishes this work conceptually or experimentally from prior studies like Chen et al. [1] and Nikulin et al. [2]?
> >
>
> Chen et al. results do not guarantee that promptable representations zero-shot satisfy properties required for Latent Action Models, since, unlike in their setup, we cannot learn representation filtering from real actions. Our exhaustive benchmarking shows that VLMs differ substantially in these capabilities, which was not previously known. And this is significant: choosing the right VLM turns LAM from non-functional into a method that performs comparably to distractor-free data, effectively overcoming LAM’s core limitations.
>
> > an the authors provide analysis or evidence clarifying why promptable representations improve latent action learning performance beyond empirical gains?
> >
>
> Thank you for this valuable concern. LAM and learning in the presence of distractors in general have been studied quite well from a theoretical point of view, by Zhang C. et al. (2025), Lachapelle S. (2025) for LAMs, and by Misra D. et al. (2024), Lamb A. et al. (2022), Levine A. et al. (2024) for representation learning. These works provide theoretical grounding and guide our empirical investigations, as we briefly discuss in Section 2.
>
> In particular, both Zhang C. et al (2025) and Misra D. et al (2024) theoretically show that latent actions can capture exogenous noise instead of actions, if the former is more predictive of changes in the observations. Moreover, Zhang C. et al shows that in linear case, LAM is equivalent to performing PCA on the mixture of controllable changes and exogenous noise, which implies that for successful action recovery, controllable changes should explain dynamics better that exogenous noise (e.g. explain more variance). This is exactly the motivation behind our work.
>
> To summarize, our work provides empirical confirmation of an already established theory.

---

> ### Author Response · Authors · 2025-11-28
> **References**
>
> References:
>
> 1. Zhang, C., Pearce, T., Zhang, P., Wang, K., Chen, X., Shen, W., ... & Bian, J. (2025). What Do Latent Action Models Actually Learn?. *arXiv preprint arXiv:2506.15691*.
> 2. Huang, H., Liu, F., Fu, L., Wu, T., Mukadam, M., Malik, J., ... & Abbeel, P. (2025). Otter: A vision-language-action model with text-aware visual feature extraction. *arXiv preprint arXiv:2503.03734*.
> 3. Lachapelle, S. (2025). On the Identifiability of Latent Action Policies. *arXiv preprint arXiv:2510.01337*.
> 4. Misra, D., Saran, A., Xie, T., Lamb, A., & Langford, J. (2024). Towards principled representation learning from videos for reinforcement learning. *arXiv preprint arXiv:2403.13765*.
> 5. Lamb, A., Islam, R., Efroni, Y., Didolkar, A., Misra, D., Foster, D., ... & Langford, J. (2022). Guaranteed discovery of control-endogenous latent states with multi-step inverse models. *arXiv preprint arXiv:2207.08229*.
> 6. Levine, A., Stone, P., & Zhang, A. (2024). Multistep inverse is not all you need. *arXiv preprint arXiv:2403.11940*.

---

### Official Review · Reviewer_iqTp · 2025-10-31

**Soundness:** 2
**Presentation:** 2
**Contribution:** 2
**Rating:** 4
**Confidence:** 4

**Summary:**

In this work, the authors tackle the problem of Latent Action Models capturing noise in noisy environments. Instead of learning the actions related to an agent's movements, the latent actions can focus too much on background noise, rendering them unusable for downstream tasks. To alleviate this, the authors propose to use a VLM to extract the relevant information from the video and use it as a target for LAM training. Exhaustive experiments on VLMs show an increase in performance of the model when distractors are present.

**Strengths:**

- This work tackles an important problem of disambiguating actions in LAM. Often, considered robotics setups are too simplistic for it to matter, but this is crucial in more complex environments.

- The authors present a clear experiment to demonstrate the targeted problem.

- The experiments across VLMs are exhaustive, both across models and prompt choices.

- Most of the performance compared to the distractor free setting is able to be recovered by the model.

**Weaknesses:**

1) Saying that the method is “without any supervision” (line 57) is slightly misleading as it relies both on task information and the use of a VLM to extract relevant information. It is however a weaker supervision than previous works using action labels.

2) As pointed out by the authors, the linear probing evaluation does not guarantee the minimality of actions and one could imagine that both the robot action as well as the “noise” are captured. While the proposed bottleneck to 128 dimensions may help (even if using a continuous space), this would need to be measured in some capacity.

3) Section 4: Observations with distractors are used as input to the IDM and FDM. But this means that the FDM has to go from a noisy observation to a clean one. In that case either the latent actions can encode this “noise removal”, or the FDM must focus only on relevant part of the data and filter the noise itself (which may be aided by using the promptable representations). The former is obviously undesirable but this is never discussed or analyzed in the paper.

4) How promptable representations are used should be made clearer. Right now the paper assumes a lot of knowledge of [1]. Perhaps a figure describing the whole process would help make this clearer.

5) Lack of formal comparison to other approaches such as UniVLA. Even if the UniVLA pipeline is more complex, it makes similar assumptions (access to task information) and should be compared to the proposed approach.

6) Why were 128 unconstrained dimensions chosen for the latent? While the VQ used by LAPO has limitations, using it (or another kind of regularization) would help with minimality of latent actions. This minimality can also have an impact on how noise is captured, as it requires significantly more capacity to learn than the robot actions that are common across videos. A proper comparison of behaviors under different latent constraints would help understand the problem better.


[1] Chen, William, et al. "Vision-language models provide promptable representations for reinforcement learning." arXiv preprint arXiv:2402.02651 (2024).

**Questions:**

1) Does the baseline LAPO also use the 128 continuous dimensions ? or is it using the original VQ ?

2) How much do you think that training in pixel space or with a small encoder is making the problem of distractors more prevalent ?

3) If promptable representations are better at filtering out noise, could they be used for everything? The VLM could be used as a general encoder for both the IDM, FDM, and FDM target


Typo line 309 : conveneince -> convenience

---

> ### Author Response · Authors · 2025-11-28
> **Official Comment by Authors (1/2)**
>
> We thank the reviewer for their feedback and for carefully engaging with our work. We apologise for the delayed response; we experienced issues accessing computing resources.
>
> We now answer the following concerns raised in the review.
>
> > Saying that the method is “without any supervision” (line 57) is slightly misleading as it relies both on task information and the use of a VLM to extract relevant information. It is however a weaker supervision than previous works using action labels.
> >
>
> Thank you for pointing that out, you are right. We have corrected it in the updated text.
>
> > As pointed out by the authors, the linear probing evaluation does not guarantee the minimality of actions and one could imagine that both the robot action as well as the “noise” are captured. While the proposed bottleneck to 128 dimensions may help (even if using a continuous space), this would need to be measured in some capacity.
> >
>
> This is a valid concern. Unfortunately, there is no direct way to measure the amount of noise in latent actions (unlike a simple probe for actual actions). The most effective way is to simply train the BC agent on the resulting actions and test it in a real environment. If the actions contain a lot of noise, they will be useless for pre-training. Since in our case we get large increases in the success rate (see Figure 9), we can confidently say that their quality is significantly improved, which in turn leads to the conclusion that there is less noise.
>
> > Section 4: Observations with distractors are used as input to the IDM and FDM. But this means that the FDM has to go from a noisy observation to a clean one. In that case either the latent actions can encode this “noise removal”, or the FDM must focus only on relevant part of the data and filter the noise itself (which may be aided by using the promptable representations). The former is obviously undesirable but this is never discussed or analyzed in the paper.
> >
>
> Thank you for pointing that out, we hadn't thought of that! That could indeed happen, but it would have a direct impact on latent actions quality metrics and the final success rate. In general, encoding ‘noise removal’ into latent actions is no better than encoding noise (and actually correlated with it) in terms of pre-training the BC on these actions, as they do not provide useful priors. Since we see a clear improvement in the action probes, as well as a significant increase in the final success rate, we can safely conclude that latent actions trained with VLMs actually filter out distractors.
>
> > How promptable representations are used should be made clearer. Right now the paper assumes a lot of knowledge of [1]. Perhaps a figure describing the whole process would help make this clearer.
> >
>
> This is an excellent suggestion, and we are actively working on it. We hope this does not hinder the reviewers’ understanding of the paper’s contributions. We aim to include the visualization by the end of the rebuttal and in the camera-ready version.
>
> We added more details about the extraction of promptable representations and moved the corresponding paragraph to Section 3 to introduce them earlier. These representations replace the next-observation target during FDM training in LAPO, which constitutes only a minimal modification.
>
> > Lack of formal comparison to other approaches such as UniVLA. Even if the UniVLA pipeline is more complex, it makes similar assumptions (access to task information) and should be compared to the proposed approach.
> >
>
> We apologise if this was not clear from the text, but UniVLA is not applicable in our setup. UniVLA simply adds the language task instruction to the IDM and FDM as input, which may help disentangle task-relevant videos from noise at a high level, but does not provide the per-step learning signal needed to accurately recover low-level ground-truth actions. Thus, during single task learning, task instruction is always a constant vector, and UniVLA converts to just LAPO. This is the general limitation of UniVLA, while we show that promptable representations are applicable even in the single task learning regime.
>
> UniVLA additionally claims that learning LAMs in DINOv2 latent space further enhances “object-centric and spatially aware properties”. We benchmarked the DINOv2 representations (see updated Figure 4), and showed that it achieves worst latent action quality among all considered methods.
>
> We added OTTER [5] as an additional baseline (see updated Figure 4 and Figure 9), as it is the closest method to ours in a sense that is explicitly constructs language conditioned representations to focus on task relevant details. The results show that representations obtained from VLM are better suited for training LAM, as they have higher success rates, 0.39 vs 0.6.

---

> ### Author Response · Authors · 2025-11-28
> **Official Comment by Authors (2/2)**
>
> > Why were 128 unconstrained dimensions chosen for the latent? ... A proper comparison of behaviors under different latent constraints would help understand the problem better.
> >
> Thank you for such an interesting and important question. There are no particular reasons for choosing 128, apart from historical consistency (with the original LAPO or LAOM). It was important for us to simply ensure that all methods under consideration had the same latent action capacity constraints. We do not use VQ/FSQ or any other quantization methods due the existing evidence [1, 2, 3, 4] that, at least for continuous action spaces, it reduces the performance and prone to codebook collapse.
>
> To provide additional comparison, we re-run the main experiments with varying latent action dimensions, from 16 up to original 128, on full datasets from all environments and with 3 random seeds. We report success rate IQM and 95% CI. Results provided below (and in updated Section 6, line 424):
>
> | Method \ Latent action dim | 16 | 32 | 64 | 128 |
> | --- | --- | --- | --- | --- |
> | LAPO wo/ distractors | 0.49 (0.44, 0.55) | 0.52 (0.45, 0.6) | 0.58 (0.53, 0.63) | 0.75 (0.73, 0.78) |
> | LAPO+Molmo wo/ distractors | **0.72** (0.71, 0.73) | **0.72** (0.71, 0.74) | **0.72** (0.7, 74) | **0.71** (0.68, 0.73) |
> | LAPO w/ distractors | 0.03 (0.02, 0.05) | 0.05 (0.04, 0.06) | 0.05 (0.03, 0.05) | 0.1 (0.08, 0.12) |
> | LAPO+Molmo w/ distractors | **0.33** (0.31, 0.36) | **0.5** (0.49, 0.55) | **0.57** (0.54, 0.6) | **0.62** (0.6, 0.63) |
>
> Interestingly, we observe that promptable representations not only allow to increase success rates in the presence of distractors, but significantly improve upon LAPO without distractors with larger minimality constraints (e.g. only 16 action dim). This further confirms that VLMs help filter out information not relevant to the controllable changes (e.g. static background).
>
> ### Questions
>
> > Does the baseline LAPO also use the 128 continuous dimensions ? or is it using the original VQ ?
> >
>
> Yes, we unify all the hyperparameters and architectures as much as possible. So, baseline LAPO uses 128 continuous dimensions, and overall a roughly the same number of parameters (both in IDM and FDM).
>
> > How much do you think that training in pixel space or with a small encoder is making the problem of distractors more prevalent ?
> >
>
> As Zhang C. et al (2025) and Misra D. et al (2024) theoretically show, the limitations of LAMs in the presence of distractors are general and arise from the properties of the data, not the specific architecture or the networks scale. Thus, it is generally does not matter. For example, LAOM [1] predicts next observations in a latent space, but still does not work without supervision. Our work specifically changes the properties of the training data, by providing cleaner targets, and not by improving the architecture or losses.
>
> > If promptable representations are better at filtering out noise, could they be used for everything? The VLM could be used as a general encoder for both the IDM, FDM, and FDM target
> >
>
> Yes, this is might be the future of LAMs! However, in our single task setup it will not provide much benefits, but will be indispensable for multi-task large scale pre-training. By training LAMs in the space of promptable representations we can get a foundation LAM model, trained not to recover single action space, but many specified by the language prompt to the VLM, e.g. zero-shot predict actions for different agents in the single video, without re-training.
>
> However, for IDM, it is important to also have access to the original observations so as not to lose any visual details. Unfortunately, VLM are known to focus too little attention on small details or visual input in general. But this will most likely improve in the future.
>
> **References**
>
> 1. Nikulin, A., Zisman, I., Tarasov, D., Lyubaykin, N., Polubarov, A., Kiselev, I., & Kurenkov, V. (2025). Latent action learning requires supervision in the presence of distractors. *arXiv preprint arXiv:2502.00379*.
> 2. Liang, A., Czempin, P., Hong, M., Zhou, Y., Biyik, E., & Tu, S. (2025). Clam: Continuous latent action models for robot learning from unlabeled demonstrations. *arXiv preprint arXiv:2505.04999*.
> 3. Yang, J., Shi, Y., Zhu, H., Liu, M., Ma, K., Wang, Y., ... & Wang, L. (2025). CoMo: Learning Continuous Latent Motion from Internet Videos for Scalable Robot Learning. *arXiv preprint arXiv:2505.17006*.
> 4. Cui, Z., Pan, H., Iyer, A., Haldar, S., & Pinto, L. (2024). Dynamo: In-domain dynamics pretraining for visuo-motor control. *Advances in Neural Information Processing Systems*, *37*, 33933-33961.
> 5. Huang, H., Liu, F., Fu, L., Wu, T., Mukadam, M., Malik, J., ... & Abbeel, P. (2025). Otter: A vision-language-action model with text-aware visual feature extraction. *arXiv preprint arXiv:2503.03734*.

---

### Official Review · Reviewer_7mj1 · 2025-11-01

**Soundness:** 3
**Presentation:** 3
**Contribution:** 3
**Rating:** 6
**Confidence:** 3

**Summary:**

This paper proposes using promptable representations from Vision-Language Models (VLMs) as a clean training target for Latent Action Models (LAMs), enabling them to learn task-centric latent actions from videos containing action-correlated distractors, without requiring ground-truth action labels.

**Strengths:**

+ Well-Motivated Solution: Directly addresses a known failure mode of LAMs.

+ The "twin observation" experiment is a powerful motivator, and the large-scale VLM benchmark provides strong, data-driven insights.

+ Experimental results demonstrates a substantial improvement in success rates on the Distracting MetaWorld benchmark.

**Weaknesses:**

- The core concept of using VLM embeddings as representations for control was previously explored by Chen et al. [9] and Huang et al. [24]. This work applies a similar idea to a different, albeit important, problem (LAM robustness).

- The benchmarking is almost exclusively against the baseline LAPO [41] and its own variants. It lacks comparison to other contemporary LAMs (e.g., UniVLA [8]) or alternative distractor-handling techniques, limiting the claim to a broader state-of-the-art.

- The analysis is heavily empirical and lacks theoretical grounding. It lacks a theoretical framework or in-depth analysis (e.g., of the embedding space) to explain why the method works beyond the initial "twin" experiment.

**Questions:**

- How does the method compare to other proposed solutions for robust LAM training, such as UniVLA, on larger-scale benchmarks?

- Beyond pre-training data, what specific properties of the VLM's representation space are critical for its success, and can this be analyzed more formally?

---

> ### Author Response · Authors · 2025-11-28
> **Official Comment by Authors (1/2)**
>
> We thank the reviewer for their thoughtful and constructive feedback and apologise for the delayed response; we experienced issues accessing computing resources. We address the main concerns below.
>
> > The core concept of using VLM embeddings as representations for control was previously explored by Chen et al. [9] and Huang et al. [24].
> >
>
> We respectfully disagree. Although the components we use are not new, this does not diminish the novelty of our work. We believe novelty should be judged not by the method alone but by the new insights and results it enables.
>
> Thus, while Chen et al. and Huang et al. also use promptable representations, their improvements are in standard RL, modest and do not transform a non-functional method into one that fully recovers performance. More importantly, their results do not guarantee the existence of zero-shot properties required for Latent Action Models, since, unlike in their setup, we cannot learn representation filtering from real actions.
>
> For LAMs, representations must (1) contain task centric visual information, (2) be minimal by filtering out visual details irrelevant to the prompt, and (3) remain consistent across dynamics to mimic changes caused by real actions. Our exhaustive benchmarking shows that VLMs differ substantially in these capabilities, which was not previously known. And this is significant: choosing the right VLM turns LAM from non-functional into a method that performs comparably to distractor-free data, effectively overcoming LAM’s core limitations.
>
> > The benchmarking is almost exclusively against the baseline LAPO [41] and its own variants. It lacks comparison to other contemporary LAMs (e.g., UniVLA [8]) or alternative distractor-handling techniques
> >
>
> We compare only to LAPO intentionally, since nearly all recent LAMs (LAOM, DynaMo, Moto, IGOR, LAPO, LAPA, Genie, GR00T N1) are mathematically equivalent to it, differing only in architectural details, but not objectives, and therefore sharing the same fundamental limitations. Moreover, the theoretical analysis of LAMs by Zhang C. et al. (2025) also considers only LAPO, as it is the simplest formulation that generalises the rest. Given that we are not claiming state-of-the-art for VLA, but rather investigating the limitations of LAM under controlled conditions, we believe that this is sufficient.
>
> We added OTTER (Huang et al. 2025) as an additional baseline (see updated Figure 4), as it is the closest method to ours in a sense that is explicitly constructs language conditioned representations to focus on task relevant details. The results show that representations obtained from VLM are better suited for training LAM, as they have higher success rates, e.g. 0.39 vs 0.6 (see updated Figure 9). We discuss UniVLA a bit later.
>
> As far as distractor-handling techniques, to the best of our knowledge, apart from our work, there are no methods aimed at specifically filtering distractors during LAM training, or in general without requirements for the existence of action or reward labels. Some techniques, such as augmentations, temporal consistency loss or avoiding reconstructing observations in image space were tried for LAMs in previous work by Nikulin A. et al. (2025) with negative results. Therefore, we do not compare ourselves to those, furthermore, they are orthogonal to our approach and can be used in conjunction (e.g., augmentation). We refer to the Appendix B in Nikulin et al. (2025) for a more detailed overview of such methods.
>
> > The analysis is heavily empirical and lacks theoretical grounding. It lacks a theoretical framework or in-depth analysis (e.g., of the embedding space) to explain why the method works beyond the initial "twin" experiment.
> >
>
> Thank you for this valuable concern. We agree that theoretical analysis is important and helps in understanding the limitations of the methods being studied. However, LAM and learning in the presence of distractors in general have been studied quite well from a theoretical point of view, by Zhang C. et al. (2025), Lachapelle S. (2025) for LAMs, and by Misra D. et al. (2024), Lamb A. et al. (2022), Levine A. et al.(2024) for representation learning. These works provide theoretical grounding and guide our empirical investigations, as we briefly discuss in Section 2.
>
> In particular, both Zhang C. et al (2025) and Misra D. et al. (2024) theoretically show that latent actions can capture exogenous noise instead of actions, if the former is more predictive of changes in the observations. Moreover, Zhang C. et al shows that in linear case, LAM is equivalent to performing PCA on the mixture of controllable changes and exogenous noise, which implies that for successful action recovery, controllable changes should explain dynamics better that exogenous noise (e.g. explain more variance). This is exactly the motivation behind our work.
>
> To summarize, our work provides additional empirical confirmation of an already established theory.

---

> ### Author Response · Authors · 2025-11-28
> **Official Comment by Authors (2/2)**
>
> ### Questions
>
> > How does the method compare to other proposed solutions for robust LAM training, such as UniVLA, on larger-scale benchmarks?
> >
>
> We apologise if this was not clear from the text, but UniVLA is not applicable in our setup. UniVLA simply adds the language task instruction to the IDM and FDM as input, which may help to disentangle task relevant videos from noise, but does not provide per-step learning signal, which is needed to truly recover low-level ground-truth actions. Thus, during single task learning, task instruction is always a constant vector, and UniVLA converts to just LAPO. This is the general limitation of UniVLA, while we show that promptable representations are applicable even in the single task learning regime.
>
> UniVLA additionally claims that learning LAMs in DINOv2 latent space further enhances “object-centric and spatially aware properties”. We benchmarked the DINOv2 representations (see updated Figure 4), and showed that it achieves worst latent action quality among all considered methods.
>
> > Beyond pre-training data, what specific properties of the VLM's representation space are critical for its success, and can this be analyzed more formally?
> >
>
> As we discussed above, on the high level it should(1) contain task centric visual information, (2) be minimal by filtering out visual details irrelevant to the prompt, and (3) remain consistent across dynamics to mimic changes caused by real actions
>
> Unfortunately, theoretical analysis of VLMs is not currently feasible. The interpretability of large LLM/VLM models is a separate large field, and precisely because of the impossibility of complete theoretical analysis, researchers resort to tools such as probing, SAE, or transcoders. Following best practices, we also use linear probing as a proxy to access the quality of the representations.
>
> References:
>
> 1. Zhang, C., Pearce, T., Zhang, P., Wang, K., Chen, X., Shen, W., ... & Bian, J. (2025). What Do Latent Action Models Actually Learn?. *arXiv preprint arXiv:2506.15691*.
> 2. Huang, H., Liu, F., Fu, L., Wu, T., Mukadam, M., Malik, J., ... & Abbeel, P. (2025). Otter: A vision-language-action model with text-aware visual feature extraction. *arXiv preprint arXiv:2503.03734*.
> 3. Nikulin, A., Zisman, I., Tarasov, D., Lyubaykin, N., Polubarov, A., Kiselev, I., & Kurenkov, V. (2025). Latent action learning requires supervision in the presence of distractors. *arXiv preprint arXiv:2502.00379*.
> 4. Lachapelle, S. (2025). On the Identifiability of Latent Action Policies. *arXiv preprint arXiv:2510.01337*.
> 5. Misra, D., Saran, A., Xie, T., Lamb, A., & Langford, J. (2024). Towards principled representation learning from videos for reinforcement learning. *arXiv preprint arXiv:2403.13765*.
> 6. Lamb, A., Islam, R., Efroni, Y., Didolkar, A., Misra, D., Foster, D., ... & Langford, J. (2022). Guaranteed discovery of control-endogenous latent states with multi-step inverse models. *arXiv preprint arXiv:2207.08229*.
> 7. Levine, A., Stone, P., & Zhang, A. (2024). Multistep inverse is not all you need. *arXiv preprint arXiv:2403.11940*.

---

### Author Response · Authors · 2025-12-01
**Summary of Our Rebuttal and Discussions**

We genuinely thank all reviewers and ACs for their efforts and time in reviewing our paper, as well as their constructive suggestions that contribute to the improvement of our work. We sincerely appreciate the positive evaluations from the reviewers.

Reviewers consistently recognized several strengths of our paper: (1) directly addressing an important and known failure mode of current Latent Actions Models (LAMs), (2) well motivated and interesting solution, leveraging  common-sense reasoning capabilities of VLMs to learn stronger latent actions centered on controllable changes, and (3) exhaustive experimental results demonstrating a substantial improvement in latent action quality and success rates on the Distracting MetaWorld benchmark.

Reviewers had several concerns, which we tried to address during the rebuttal with the additional experiments and clarifications.

1. **Novelty concern.** Indeed, promptable representation were previously explored by Chen et al. for RL. We clarified that our contribution is ***new setting and novel empirical insights***: (1) results from Chen et al does not guarantee success in latent action learning, (2) we show that VLMs differ dramatically in zero-shot suitability for LAMs, and the *right* VLM makes LAM functional under distractors, increasing success rates sixfold, something not shown in prior work.
2. **Lack of baselines.** We added OTTER (Huang et al. 2025) as an additional baseline (see updated Figure 4), as it is the closest method to ours in a sense that is explicitly constructs language conditioned representations to focus on task relevant details. The results show that representations obtained from VLM are better suited for training LAM, as they have higher success rates, e.g. 0.39 vs 0.6 (see updated Figure 9). In addition, we explained why **UniVLA is not applicable**: in single-task settings its language input is constant, giving no per-step learning signal, reducing it to simply LAPO. We revised the related work section to better highlight this limitation.
3. **Comparison with the self-supervised learning representations.** We provided additional results with representations obtained from DINOv2 and CLIP (see updated Figure 4 and line 312). Both DINOv2 and CLIP representations achieved worst latent action quality among all considered methods. Without language-conditioning there is no guarantee that SSL methods focus on exactly what is controllable in the task. To demonstrate this even more clearly, we compared the representations from CLIP and OTTER (see updated Figure 4). OTTER uses the same CLIP model, but adds simple training-free filtering using text CLIP embeddings. This small change significantly improves latent action quality, showing that language-conditioning is vital.
4. **Comparison under different latent constraints.** We re-run the main experiments with varying latent action dimensions. We report success rate IQM and 95% CI. Results provided below (and in updated Section 6, line 424):


    | Method \ Latent action dim | 16 | 32 | 64 | 128 |
    | --- | --- | --- | --- | --- |
    | LAPO wo/ distractors | 0.49 (0.44, 0.55) | 0.52 (0.45, 0.6) | 0.58 (0.53, 0.63) | 0.75 (0.73, 0.78) |
    | LAPO+Molmo wo/ distractors | **0.72** (0.71, 0.73) | **0.72** (0.71, 0.74) | **0.72** (0.7, 74) | **0.71** (0.68, 0.73) |
    | LAPO w/ distractors | 0.03 (0.02, 0.05) | 0.05 (0.04, 0.06) | 0.05 (0.03, 0.05) | 0.1 (0.08, 0.12) |
    | LAPO+Molmo w/ distractors | **0.33** (0.31, 0.36) | **0.5** (0.49, 0.55) | **0.57** (0.54, 0.6) | **0.62** (0.6, 0.63) |

    We observe that promptable representations not only allow to increase success rates in the presence of distractors, but significantly improve upon LAPO without distractors with larger minimality constraints (e.g. only 16 action dim). This further confirms that VLMs help filter out information not relevant to the controllable changes.

5. **Harder distractors.** We added a change in lighting throughout the episode (see video examples in the updated supplementary materials), in addition to the videos. Due to time constraints, we chose one task (button-press-topdown), but for the camera-ready version, we will add up the results for the entire MT10. We used 3 random seeds and full datasets with 5k trajectories. We summarize results in a table below:


    | Method | Success rate |
    | --- | --- |
    | LAPO wo/ distractors | 0.99 |
    | LAPO+Molmo wo/ distractors | 0.96 |
    | LAPO w/ distractors | 0.21 |
    | LAPO+Molmo w/ distractors | **0.85** |

All additional experiments and discussions will be incorporated into the final version. Additionally, all already incorporated changes are highlighted via blue color in the text.

Best,

Authors

---

### Meta-Review · Area_Chair_3pjT · 2026-01-08

**Summary:**

This paper proposed to use promptable representation from VLM to help learn latent action models. The authors claim that the common-sense reasoning ability can help separate controllable changes from noise and lead to latent action models that is robust to background noise. A comprehensive experiments on both clean and noisy simulation data demonstrate the effectiveness of the proposed mehtods.

Reviewers acknowledge that the proposed method is well motivated and tackles an important problem. The authors did extensive experiments to demonstrate the proposed method consistently improves the performance compared with baselines for both clean and noisy data. However, as raised by the reviewers, this paper is only evaluated on synthetic data, no real data are used for benchmarking. The distractor (noise) added is too artificial and is not convincing. Reviewers also raised concerns that the paper needs to be improved. Considering these concerns, I tend to reject this paper.

**Reviewer Concerns:**

Concerns that are not well addressed and informed me to suggest to reject this paper are:
1. No experiments showing the method are effective on real world models (by reviewer 8hu2). The authors did not give a good answer to this. Especially, both LAPO and OTTER used real videos as benchmark in their experiments.
2. Distractor (noise) used is too artificial (raised by reviewer mdb4). The authors try to include lighting changes to address this concern, however, this is not enough. Lighting changes in a simulation environment can still be too artificial. Using real work videos (as the first concern mentioned) would be more convincing.
3. The paper could be improved (raised by mdb4). The paper is not well structured and needs a lot of work.
4. Several reviewers require the authors to compare with UniVLA. The authors claimed that this is not applicable. However, I feel the reason the authors give is a bit confuse to me and is not convincing.

**Reviewer Scores:**

The paper originally get scores of 4, 4, 4, 6. Considering some of the concerns are not well addressed, I think reviewers may remain their scores.

---

### Decision · Program_Chairs · 2026-01-26

Reject